



**Assessing the impact of anthropogenic pollution on isoprene-derived secondary organic**
**aerosol formation in PM$_{2.5}$ collected from the Birmingham, Alabama ground site during the**
**2013 Southern Oxidant and Aerosol Study**
W. Rattanavaraha[1], K. Chu[1], S. H. Budisulistiorini[1,a], M. Riva[1], Y.-H. Lin[1,b], E. S. Edgerton[2], K.
Baumann[2], S. L. Shaw[3], H. Guo[4], L. King[4], R. J. Weber[4], E. A. Stone[5], M. E. Neff[5], J. H.
Offenberg[6], Z. Zhang[1], A. Gold[1], and J. D. Surratt[1,*]
[1] Department of Environmental Sciences and Engineering, Gillings School of Global Public
Health, The University of North Carolina at Chapel Hill, Chapel Hill, NC, USA
[2] Atmospheric Research & Analysis, Inc., Cary, NC, USA
[3] Electric Power Research Institute, Palo Alto, CA, USA
[4] Earth and Atmospheric Science, Georgia Institute of Technology, Atlanta, GA, USA
[5] Department of Chemistry, University of Iowa, Iowa City, IA, USA
[6] Human Exposure and Atmospheric Sciences Division, United States Environmental Protection
Agency, Research Triangle Park, NC, USA
[a] now at: Earth Observatory of Singapore, Nanyang Technological University, Singapore
[b] now at: Michigan Society of Fellows, Department of Chemistry, University of Michigan, Ann
Arbor, MI, USA
* To whom correspondence should be addressed. Email: surratt@unc.edu
For Submission to: Atmospheric Chemistry & Physics Discussions





## Abstract

In the southeastern U.S., substantial emissions of isoprene from deciduous trees undergo
atmospheric oxidation to form secondary organic aerosol (SOA) that contributes to fine particulate
matter ($PM_{2.5}$). Laboratory studies have revealed that anthropogenic pollutants, such as sulfur
dioxide ($SO_2$), oxides of nitrogen ($NO_x$), and aerosol acidity, can enhance SOA formation from
the hydroxyl radical (OH)-initiated oxidation of isoprene; however, the mechanisms by which
specific pollutants enhance isoprene SOA in ambient $PM_{2.5}$ remain unclear. As one aspect of an
investigation to examine how anthropogenic pollutants influence isoprene-derived SOA
formation, high-volume $PM_{2.5}$ filter samples were collected at the Birmingham, Alabama (BHM)
ground site during the 2013 Southern Oxidant and Aerosol Study (SOAS). Sample extracts were
analyzed by gas chromatography/electron ionization-mass spectrometry (GC/EI-MS) with prior
trimethylsilylation and ultra performance liquid chromatography coupled to an electrospray
ionization high-resolution quadrupole time-of-flight mass spectrometry (UPLC/ESI-HR-
QTOFMS) to identify known isoprene SOA tracers. Tracers quantified using both surrogate and
authentic standards were compared with collocated gas- and particle-phase data as well as
meteorological data provided by the Southeastern Aerosol Research and Characterization
(SEARCH) network to assess the impact of anthropogenic pollution on isoprene-derived SOA
formation. Results of this study reveal that isoprene-derived SOA tracers contribute a substantial
mass fraction of organic matter (OM) (~7 to ~20%). Isoprene-derived SOA tracers correlated with
sulfate ($SO_4^{2-}$) ($r^2 = 0.34$, n = 117), but not with $NO_x$. Moderate correlation between methacrylic
acid epoxide and hydroxymethyl-methyl-$\alpha$-lactone (MAE/HMML)-derived SOA tracers and
nitrate radical production ($P[NO_3]$) ($r^2 = 0.57$, n = 40) were observed during nighttime, suggesting
a potential role of $NO_3$ radical in forming this SOA type. However, the nighttime correlation of





these tracers with nitrogen dioxide ($NO_2$) ($r^2 = 0.26$, n = 40) was weaker. Ozone ($O_3$) correlated
strongly with MAE/HMML-derived tracers ($r^2 = 0.72$, n = 30) and moderately with 2-methyltetrols
($r^2 = 0.34$, n = 15) during daytime only, suggesting that a fraction of SOA formation could occur
from isoprene ozonolysis in urban areas. No correlation was observed between aerosol pH and
isoprene-derived SOA. Lack of correlation between aerosol acidity and isoprene-derived SOA
indicates that acidity is not a limiting factor for isoprene SOA formation at the BHM site as
aerosols were acidic enough to promote multiphase chemistry of isoprene-derived epoxides
throughout the duration of the study. All in all, these results confirm the reports that anthropogenic
pollutants enhance isoprene-derived SOA formation.



## 1. Introduction


Fine particulate matter, suspensions of liquid or solid aerosol in a gaseous medium that are
less than or equal to 2.5 μm in diameter ($PM_{2.5}$), play a key role in physical and chemical
atmospheric processes. They influence climate patterns both directly, through the absorption and
scattering of solar and terrestrial radiation, and indirectly, through cloud formation (Kanakidou et
al., 2005). In addition to climatic effects, $PM_{2.5}$ has been demonstrated to pose a potential human
health risk through inhalation exposure (Pope and Dockery, 2006; Hallquist et al., 2009). Despite
the strong association of $PM_{2.5}$ with climate change and environmental health, there remains a need
to more fully resolve its composition, sources, and chemical formation processes in order to
develop effective control strategies to address potential hazards in a cost-effective manner
(Hallquist et al., 2009; Boucher et al., 2013; Nozière et al., 2015).
Atmospheric $PM_{2.5}$ are comprised in a large part (up to 90% by mass in some locations),
of organic matter (OM) (Carlton et al., 2009; Hallquist et al., 2009). OM can be derived from many
sources. Primary organic aerosol (POA) is emitted from both natural (e.g., fungal spores,
vegetation, vegetative detritus) and anthropogenic sources (fossil fuel and biomass burning) prior
to atmospheric processing. As a result of large anthropogenic sources, POA is abundant largely in
urban areas. Processes such as biomass burning and combustion also yield volatile organic
compounds (VOCs), which have high vapor pressures and can undergo atmospheric oxidation to
form secondary organic aerosol (SOA) through gas-to-particle phase partitioning (condensation or
nucleation) with subsequent particle-phase (multiphase) chemical reactions (Grieshop et al.,

2009).

At around 600 Tg emitted per year into the atmosphere, isoprene (2-methyl-1,3-butadiene,
$C_5H_8$) is the most abundant volatile non-methane hydrocarbon (Guenther et al., 2012). The



abundance of isoprene is particularly high in the southeastern U.S. due to emissions from broadleaf
deciduous tree species (Guenther et al., 2006). Research over the last decade has revealed that
isoprene, via hydroxyl radical (OH)-initiated oxidation, is a major source of SOA (Claeys et al.,
2004; Edney et al., 2005; Kroll et al., 2005 ; Kroll et al., 2006; Surratt et al., 2006;  Lin et al., 2012;
Lin et al., 2013a). In addition, it is known that SOA formation is enhanced by anthropogenic
emissions, namely oxides of nitrogen ($NO_x$) and sulfur dioxide ($SO_2$), that are a source of acidic
aerosol onto which photochemical oxidation products of isoprene are reactively taken up to yield
a variety of SOA products (Edney et al., 2005; Kroll et al., 2006; Surratt et al., 2006; Surratt et al.,
2007b; Surratt et al., 2010;  Lin et al., 2013b;) .

Recent work has begun to elucidate some of the critical intermediates of isoprene oxidation

that lead to SOA formation through acid-catalyzed heterogeneous chemistry (Kroll et al., 2005;
Surratt et al., 2006). Under low-$NO_x$ conditions, such as in a pristine environment, isomeric
isoprene epoxydiols (IEPOX) have been demonstrated to be critical to the formation of isoprene
SOA. On advection of IEPOX to an urban environment and mixing with anthropogenic emissions
of acidic sulfate aerosol, SOA formation is enhanced (Surratt et al., 2006; Lin et al., 2012; Lin et
al., 2013b). This pathway has been shown to yield 2-methyltetrols as major SOA constituents of
ambient $PM_{2.5}$ (Claeys et al, 2004; Surratt et al., 2010; Lin et al., 2012). Further work has revealed
a number of additional IEPOX-derived SOA tracers, including $C_5$-alkene triols (Wang et al., 2005;
Lin et al., 2012), *cis*- and *trans*-3-methyltetrahydrofuran-3,4-diols (3-MeTHF-3,4-diols) (Lin et
al., 2012; Zhang et al., 2012), IEPOX-derived organosulfates (OSs) (Lin et al., 2012), and IEPOX-
derived oligomers (Lin et al., 2014). Some of the IEPOX-derived oligomers have been shown to
contribute to aerosol components known as brown carbon that absorb light in the near ultraviolet
(UV) and visible ranges (Lin et al., 2014). Under high-$NO_x$ conditions, such as encountered in an





urban environment, isoprene is oxidized to methacrolein and SOA formation occurs via the further
oxidation of methacrolein (MACR) (Kroll et al., 2006; Surratt et al., 2006) to methacryloyl
peroxynitrate (MPAN) (Chan et al., 2010; Surratt et al., 2010; Nguyen et al., 2015). It has recently
been shown that when MPAN is oxidized by OH it yields at least two SOA precursors, methacrylic
acid epoxide (MAE) and hydroxymethyl-methyl-$\alpha$-lactone (HMML) (Surratt et al., 2006; Surratt
et al., 2010; Lin et al., 2013a; Nguyen et al., 2015). Whether SOA precursors are formed under
high- or low-$NO_x$ conditions, aerosol acidity is a critical parameter that enhances the reaction
kinetics through acid-catalyzed reactive uptake and multiphase chemistry of either IEPOX or
MAE/HMML ( Surratt et al., 2007b; Surratt et al., 2010; Lin et al., 2013b).

Due to the considerable emissions of isoprene, an SOA yield of even 1% would contribute

significantly to ambient SOA (Carlton et al., 2009; Henze et al., 2009). This conclusion is
supported by measurements showing that up to a third of total fine OA mass can be attributed to
IEPOX-derived SOA tracers in Atlanta, GA (JST) during summer months (Budisulistiorini et al.,
2013; Budisulistiorini et al., 2015). A recent study in Yorkville, GA (YRK), similarly found that
IEPOX-derived SOA tracers comprised 12-19% of the fine OA mass (Lin et al., 2013b). Another
SOAS site at Centreville, Alabama (CTR) revealed IEPOX-SOA contributed 18% of total OA
mass (Xu et al., 2015). The individual ground sites corroborate recent aircraft-based measurements
made in the Studies of Emissions and Atmospheric Composition, Clouds, and Climate Coupling
by Regional Surveys (SEAC4RS) aircraft campaign, which estimates an IEPOX-SOA contribution
of 32% to OA mass in the southeastern U.S. (Hu et al., 2015).

It is clear from the field studies discussed above that particle-phase chemistry of isoprene-

derived oxidation products plays a large role in atmospheric SOA formation. However, much
remains unknown regarding the exact nature of its formation, limiting the ability of models to



accurately account for isoprene SOA (Carlton et al., 2010b; Foley et al., 2010). Currently,
traditional air quality models in the southeastern U.S. do not incorporate detailed particle-phase
chemistry of isoprene oxidation products (IEPOX or MAE/HMML) and generally under-predict
isoprene SOA formation (Carlton et al., 2010a). Recent work demonstrates that incorporating the
specific chemistry of isoprene epoxide precursors into models increases the accuracy of isoprene
SOA prediction (Pye et al., 2013; Karambelas et al., 2014), suggesting that understanding the
formation mechanisms of biogenic SOA, especially with regard to the effects of anthropogenic
emissions, such as $NO_x$ and $SO_2$, will be key to more accurate models. More accurate models are
needed in order to devise cost-effective control strategies for reducing $PM_{2.5}$ levels. Since isoprene
is primarily biogenic in origin, and therefore not controllable, the key to understanding the public
health and environmental implications of isoprene SOA lies in resolving the effects of
anthropogenic pollutants.

This study presents results from the 2013 Southeastern Oxidant and Aerosol Study

(SOAS), where several well-instrumented ground sites dispersed throughout the southeastern U.S.
made intensive gas- and particle-phase measurements from June 1 – July 16, 2013. The primary
purpose of this campaign was to examine, in greater detail, the formation mechanisms,
composition, and properties of biogenic SOA, including the effects of anthropogenic emissions.
This study pertains specifically to the results from the BHM ground site, where the city's ample
urban emissions mix with biogenic emissions from the surrounding rural areas, creating an ideal
location to investigate such interactions. The results presented here focus on analysis of $PM_{2.5}$
collected on filters during the campaign by gas chromatography interfaced to electron ionization-
mass spectrometry (GC/EI-MS) and ultra performance liquid chromatography interfaced with
electrospray ionization high-resolution quadrupole time-of-flight mass spectrometry (UPLC/ESI-



HR-QTOFMS). The analysis of $PM_{2.5}$ was conducted in order to measure quantities of known
isoprene SOA tracers and using collocated air quality and meteorological measurements to
investigate how anthropogenic pollutants including $NO_x$, $SO_2$, aerosol acidity (pH), $PM_{2.5}$ sulfate
($SO_4^{2-}$), and $O_3$ affect isoprene SOA formation. These results, along with the results presented
from similar studies during the 2013 SOAS campaign, seek to elucidate the chemical relationships
between anthropogenic emissions and isoprene SOA formation in order to provide better
parameterizations needed to improve the accuracy of air quality models in this region of the U.S.
**2.  Methods**
**2.1. Site description and collocated data**
Filter samples were collected in the summer of 2013 as part of the SOAS field campaign
at the BHM ground site (33.553N, 86.815W). In addition to the SOAS campaign, the site is also
part of the Southeastern Aerosol Research and Characterization Study (SEARCH) (Figure S1 of
the Supplement), an observation and monitoring program initiated in 1998. SEARCH and this site
are described elsewhere in detail (Hansen et al., 2003; Edgerton et al., 2006). The BHM site is
surrounded by significant transportation and industrial sources of PM. West of BHM are US-31
and I-65 highways. To the north, northeast and southwest of BHM several coking ovens and an
iron pipe foundry are located (Hansen et al., 2003).
**2.2. High-Volume filter sampling and analysis methods**
**2.2.1.  High-Volume filter sampling**
From June 1 – July 16, 2013, $PM_{2.5}$ samples were collected onto Tissuquartz™ Filters
(8 x 10 in, Pall Life Sciences) using high-volume $PM_{2.5}$ samplers (Tisch Environmental) operated
at 1 $m^3$ $min^{-1}$ at ambient temperature described in detail elsewhere (Budisulistiorini et al. 2015;



Riva et al., 2015). All quartz filters were pre-baked prior to collection. The procedure consisted of
baking filters at 550 °C for 18 hours followed by cooling to 25 °C over 12 hours.

The sampling schedule is given in Table 1. Either two or four samples were collected per

day. The regular schedule consisted of two samples per day, one during the day, the second at
night, each collected for 11 hours. On intensive sampling days, four samples were collected, with
the single daytime sample being subdivided into three separate periods. The intensive sampling
schedule was conducted on days when high levels of isoprene, $SO_4^{2-}$ and $NO_x$ where forecast by
the National Center for Atmospheric Research (NCAR) using the Flexible Particle dispersion
model (FLEXPART) (Stohl et al., 2005) and Model for Ozone and Related Chemical Tracers
(MOZART) (Emmons et al., 2010) simulations. Details of these simulations have been
summarized in Budisulistiorini et al. (2015); however, these model data were only used
qualitatively to determine the sampling schedule. The intensive collection frequency allowed
enhanced time resolution for offline analysis to examine the effect of anthropogenic emissions on
the evolution of isoprene SOA tracers throughout the day.

In total, 120 samples were collected throughout the field campaign with a field blank filter

collected every 10 days to identify errors or contamination in sample collection and analysis. All
filters were stored at -20 °C in the dark until extraction and analysis. In addition to filter sampling
of $PM_{2.5}$, SEARCH provided a suite of additional instruments at the site collecting measurements
of a variety of variables, including meteorology, gas, and continuous PM monitoring. The variables
with respective instrumentation are summarized in Table S1 of the Supplement.
**2.2.2.  Isoprene-derived SOA analysis by GC/EI-MS**



SOA collected in the field on quartz filters was extracted and isoprene tracers quantified

by GC/EI-MS with prior trimethylsilylation. A 37-mm diameter circular punch from each filter

was extracted in a pre-cleaned scintillation vial with 20 mL of high-purity methanol (LCMS

CHROMASOLV-grade, Sigma-Aldrich) by sonication for 45 minutes. The extracts were filtered

through PTFE syringe filters (Pall Life Science, Acrodisc®, 0.2-µm pore size) to remove insoluble

particles and residual quartz fibers. The filtrate was then blown dry under a gentle stream of $N_2$ at

room temperature. The dried residues were immediately trimethylsilylated by reaction with 100

µL of BSTFA + TMCS (99:1 *v/v*, Supelco) and 50 *µ*L of pyridine (anhydrous, 99.8 %, Sigma-

Aldrich) at 70 °C for 1 hour. Derivatized samples were analyzed within 24 hours after

trimethylsilylation using a Hewlett-Packard (HP) 5890 Series II Gas Chromatograph coupled to a

HP 5971A Mass Selective Detector. The gas chromatograph was equipped with an *Econo-Cap®-*

*EC®-5* Capillary Column (30 m x 0.25 mm i.d.; 0.25-µm film thickness) to separate trimethylsilyl

derivatives before MS detection. 1 *µ*L aliquots were injected onto the column. Operating

conditions and procedures have been described elsewhere (Surratt et al., 2010).

Extraction efficiency was assessed and taken into account for the quantification of all SOA

tracers. Efficiency was determined by analyzing 4 pre-baked filters spiked with 50 ppmv of 2-

methyltetrols, 2-methylglyceric acid, levoglucosan, and *cis*- and *trans*-3-MeTHF-3,4-diols.

Extraction efficiency was above 90% and used to correct the quantification of samples. Extracted

ion chromatograms (EICs) of *m/z* 262, 219, 231, 335 were used to quantify the *cis*-/*trans*-3-

MeTHF-3,4-diols, 2-methyltetrols and 2-methylglyceric acid, $C_5$-alkene triols, and IEPOX-

dimers, respectively (Surratt et al., 2006).

2-Methyltetrols were quantified using an authentic reference standard that consisted of a

mixture of racemic diasteroisomers. Similarly, 3-MeTHF-3,4-diol isomers were also quantified





using authentic standards; however, 3-MeTHF-3,4-diol isomers were detected in few field
samples. 2-Methylglyceric acid was also quantified using an authentic standard. Procedures for
synthesis of the 2-methyltetrols, 3-MeTHF-3,4-diol isomers, and 2-methylglyceric acid have been
described elsewhere (Zhang et al., 2012; Budisulistiorini et al., 2015). $C_5$-alkene triols and IEPOX-
dimers were quantified using the average response factor of the 2-methyltetrols.

### 2.2.3. Isoprene-derived SOA analysis by UPLC/ ESI-HR-QTOFMS

A 37-mm diameter circular punch from each quartz filter was extracted following the same
procedure described in section 2.2.1 for GC/EI-MS analysis. The dried residues were reconstituted
with 150 $\mu$l of a 50:50 (v/v) solvent mixture of methanol (LC-MS CHROMASOVL-grade, Sigma-
Aldrich) and high-purity water (Milli-Q, 18.2 M$\Omega$). The extracts were immediately analyzed by
the UPLC/ESI-HR-QTOFMS (6520 Series, Agilent) operated in the negative ion mode. Detailed
operating conditions have been described elsewhere (Riva et al., 2015). Mass spectra were
acquired at a mass resolution 7000-8000 over the range $m/z$ 200 – 400.
Extraction efficiency was determined by analyzing 3 pre-baked filters spiked with propyl
sulfate and octyl sulfate (electronic grade, City Chemical LLC). Extraction efficiencies were in the
range 86 – 95%. EICs of $m/z$ 215, 333 and 199 were used to quantify the IEPOX-derived OS,
IEPOX-derived dimer OS and the MAE/HMML-derived OS, respectively (Surratt et al., 2007a).
EICs were generated with a ± 5 ppm tolerance. All accurate masses for all measured
organosulfates were within ± 5 ppm. For simplicity, only the nominal masses are reported in the
text when describing these products. IEPOX-derived OS and IEPOX-derived dimer OS were
quantified by authentic standards (Zhang et al., 2012). The MAE/HMML-derived OS was
quantified using authentic MAE/HMML-derived OS synthesized in-house by a procedure to be
described in a forthcoming publication ([1]H NMR trace, Figure S2).



EICs of of *m/z* 155, 169 and 139 were used to quantify the glyoxal-derived OS,

methylglyoxal-derived OS, and the hydroxyacetone-derived OS, respectively (Surratt et al.,

2007a). In addition, EICs of *m/z* 211, 260 and 305 were used to quantify other known isoprene-

derived OSs (Surratt et al., 2007a). Glycolic acid sulfate synthesized in-house was used as a

standard to quantify the glyoxal-derived OS (Galloway et al., 2009) and propyl sulfate, was used

as a surrogate standard to quantify the remaining isoprene-derived OSs.

**2.2.4. OC and WSOC analysis**

A 1.5 $cm^2$ square punch from each quartz filter was analyzed for total organic carbon (OC)

and elemental carbon (EC) by the thermal-optical method (Birch and Cary, 1996) on a Sunset

Laboratory OC/EC instrument (Tigard, OR) at the National Exposure Research Laboratory

(NERL) at the U.S. Environmental Protection Agency, Research Triangle Park, NC. The details

of the instrument and analytical method have been described elsewhere (Birch and Cary, 1996). In

addition to the internal calibration using methane gas, four different mass concentrations of sucrose

solution were used to verify the accuracy of instrument during the analysis.

Water-soluble organic carbon (WSOC) was measured in aqueous extracts of quartz fiber filter

samples using a total organic carbon (TOC) analyzer (Sievers 5310C, GE Water & Power)

equipped with an inorganic carbon remover (Sievers 900). To maintain low background carbon

levels, all glassware used was washed with water, soaked in 10% nitric acid, and baked at 500˚C

for 5 h and 30 min prior to use. Samples were extracted in batches that consisted of 12-21 $PM_{2.5}$

samples and field blanks, one laboratory blank, and one spiked solution. A 17.3 $cm^2$ filter portion

was extracted with 15 mL of purified water (> 18 MΩ, Barnstead Easypure II, Thermo Scientific)

by ultra-sonication (Branson 5510). Extracts were then passed through a 0.45 $\mu$m PTFE filter to

remove insoluble particles. The TOC analyzer was calibrated using potassium hydrogen phthalate





(KHP, Sigma Aldrich) and was verified daily with sucrose (Sigma Aldrich). Samples and standards
were analyzed in triplicate; the reported values correspond to the average of the second and third
trials. Spiked solutions yielded recoveries that averaged (± one standard deviation) 96 ± 5 % (n =
9). All ambient concentrations were field blank subtracted.
**2.2.5. Estimation of aerosol pH by ISORROPIA**
Aerosol pH was estimated using a thermodynamic model, ISORROPIA-II (Nenes et al.,
1998). $SO_4^{2-}$, nitrate ($NO_3^-$), and ammonium ($NH_4^+$) ion concentrations measured in $PM_{2.5}$
collected from BHM, as well as relative humidity (RH), temperature and gas-phase ammonia
($NH_3$) were used as inputs into the model. These variables were obtained from the SEARCH
network at BHM, which collected the data during the period covered by the SOAS campaign. The
ISORROPIA-II model estimates particle hydronium ion concentration per unit volume of air ($H^+$,
$\mu g\ m^{-3}$), aerosol liquid water content (LWC, $\mu g\ m^{-3}$), and aqueous aerosol mass concentration ($\mu g$
$m^{-3}$). The model-estimated parameters were used in the following formula to calculate the aerosol
pH:

$$\text{Aerosol pH} = -\log_{10} a_{H^+} = -\log_{10}(\frac{H_{air}^+}{LMASS} \times \rho_{aer} \times 1000 )$$

where $a_{H^+}$ is $H^+$ activity in the aqueous phase (mol $L^{-1}$), $LMASS$ is total liquid-phase aerosol mass
($\mu g\ m^{-3}$) and $\rho_{aer}$ is aerosol density. Details of the ISORROPIA-II model and its ability to predict
pH, LWC, and gas-to-particle partitioning are not the focus of this study and are discussed
elsewhere and (Fountoukis et al., 2009).
**2.2.6. Estimation of nighttime $NO_3$**
Nitrate radical ($NO_3$) production (P[$NO_3$]) was calculated using the following equation:





$$P[NO_3] = [NO_2][O_3]k$$

where $[NO_2]$ and $[O_3]$ correspond to the measured ambient $NO_2$ and $O_3$ concentrations (mol
$cm^{-3}$), respectively, and $k$ is the temperature-dependent rate constant (Herron and Huie, 1974;
Graham and Johnston, 1978).  Since no direct measure of $NO_3$ radical was made at this site during
SOAS, $P[NO_3]$ was used as a proxy for $NO_3$ radicals present in the atmosphere to examine if there
is any association of it with isoprene-derived SOA tracers.
**3.     Results and Discussion**
**3.1.  Overview of the study**

The campaign extended from June 1 through July 16, 2013. Temperature during this period

ranged from a high of 32.6 °C to a low of 20.5 °C, with an average of ~26.4 °C. RH varied from
37-96% throughout the campaign, with an average of 71.5%. Rainfall occurred intermittently over
2-3 day periods and averaged 0.1 inches per day. Wind analysis reveals that air masses approached
largely from the south-southeast at an average wind speed of 2 m $s^{-1}$. Summaries of meteorological
conditions as well as wind speed and direction during the course of the campaign are given in
Table 2 and illustrated in Figures 1 and 2.

The average concentration of carbon monoxide (CO), a combustion byproduct, was 208.7

ppbv. The mean concentration of $O_3$ was significantly higher (*t-test, p-value* < 0.05) on intensive
sampling days (37.0 ppbv) than regular sampling days (25.2 ppbv). Concentrations of $NO_x$, $NH_3$,
and $SO_2$ were lower averaging 7.8, 1.9, and 0.9 ppbv, respectively. On average, OC and WSOC
levels were 7.2 (n = 120) and 4 $\mu$g m$^{-3}$ (n = 100), respectively. The largest inorganic component of
$PM_{2.5}$ was $SO_4^{2-}$, which averaged 2 $\mu$g m$^{-3}$ with excursions between 0.4 and 4.9 $\mu$g m$^{-3}$ during the
campaign. $NH_4^+$ and $NO_3^-$ were present at low levels, averaging 0.66 and 0.14 $\mu$g m$^{-3}$, respectively.
Time series of gas and $PM_{2.5}$ components are shown in Figure 2. WSOC accounted for 35% of OC





mass (Figure S3a), and was smaller than that recently reported in rural areas during SOAS
(Budisulistiorini et al., 2015; Hu et al., 2015), but consistent with previous observations at the
BHM site (Ding et al., 2008). WSOC/OC ratios are commonly lower in urban than rural areas, as
a consequence of higher primary OC emissions; thus, PM at BHM probably contains increased
OC.

Diurnal variation of meteorological parameters, trace gases, and $PM_{2.5}$ components are

shown in Figure S4 of the Supplement. Temperature dropped during nighttime, and reached a
maximum in the afternoon (Figure S4a). Conversely, RH was low during day and high at night.
High-$NO_x$ levels were found in the early morning and decreased during the course of the day
(Figure S4c), most likely in conjunction with rising $O_3$ levels. $O_3$ reached a maximum
concentration between 12 - 3 pm due to photochemistry (Figure S4b). $SO_2$ was slightly higher in
the morning (Figure S4c), but decreased during the day most likely as a result of planetary
boundary layer (PBL) dynamics. $NH_3$ remained fairly constant throughout the day (Figure S4c).
No significant diurnal variation was found in the concentration of inorganic $PM_{2.5}$ components,
including $SO_4^{2-}$, $NO_3^-$, and $NH_4^+$ (Figure S4d). Unfortunately, a measurement of isoprene could
not be made at BHM during the campaign. However, the diurnal trend of isoprene levels might be
similar to the data at the CTR site (Xu et al., 2015), which is only 61 miles away from BHM. Xu
et al. (2015) observed the highest levels of isoprene ($\sim$ 6 ppb) at CTR in the mid-afternoon (3 pm
local time) and its diurnal trend was similar to isoprene-OA measured by the Aerodyne Aerosol
Mass Spectrometer (AMS) during the SOAS campaign.
**3.2 Characterization of Isoprene SOA**

Table 3 summarizes the mean and maximum concentrations of known isoprene-derived

SOA tracers detected by GC/EI-MS and UPLC/ESI-HR-QTOFMS. Levoglucosan was also



analyzed as a tracer for biomass burning. Among the isoprene-derived SOA tracers, the highest
mean concentration was for 2-methyltetrols (376 ng m$^{-3}$), followed by the sum of C$_5$-alkene triols
(181 ng m$^{-3}$) and the IEPOX-derived OS (165 ng m$^{-3}$). The concentrations account for 3.8%, 1.8%
and 1.6%, respectively, of total OM mass. Noteworthy is that maximum concentrations of 2-
methylerythritol (a 2-methyltetrol isomer; 1049 ng m$^{-3}$), IEPOX-derived OS (865 ng m$^{-3}$) and (E)-
2-methylbut-3-ene-1,2,4-triol (879 ng m$^{-3}$) were attained during the intensive sampling period 4-7
pm local time on June 15, 2013, following five consecutive days of dry weather (Figure 2a and
2d) when high levels of isoprene, SO$_4^{2-}$, and NO$_x$ were forecast.

Together, the IEPOX-derived SOA tracers, which represent SOA formation from isoprene

oxidation predominantly under the low-NO$_x$ pathway, comprised 92.5% of the total detected
isoprene-derived SOA tracer mass at the BHM site. This contribution is slightly lower than
observations reported at rural sites located in Yorkville, GA (97.5%) and Look Rock, Tennessee
(LRK) (97%) (Lin et al., 2013b; Budisulistiorini et al., 2015).

The sum of MAE/HMML-OS and 2-MG, which represent SOA formation from isoprene

oxidation predominantly under the high-NO$_x$ pathway, contributed 3.25% of the total isoprene-
derived SOA tracer mass, while the OS derivative of glycolic acid (GA sulfate) contributed 3.3%.
The contribution of GA sulfate was consistent with the level of GA sulfate measured by the
airborne NOAA Particle Analysis Laser Mass Spectrometer (PALMS) over the continental U.S.
during the Deep Convective Clouds and Chemistry Experiment and SEAC4RS (Liao et al., 2015).
However, the contribution of GA sulfate to the total OM at BHM (0.3%) is lower than aircraft-
based measurements made by Liao et al. (2015) near the ground in the eastern U.S. (0.9%). GA
sulfate can form from biogenic and anthropogenic emissions other than isoprene, including





glyoxal, which is thought to be a primary source of GA sulfate (Galloway et al., 2009). For this
reason, GA sulfate will not be further discussed in this study.
Isoprene SOA contribution to total OM was estimated by assuming the OM/OC ratio 1.6
based on the recent studies (El-Zanan et al., 2009; Simon et al., 2011; Ruthenburg et al., 2014;
Blanchard et al., 2015). On average, isoprene-derived SOA tracers (sum of both IEPOX- and
MAE/HMML-derived SOA tracers) contributed ~7% (ranging to ~ 20% at times) of the total
particulate OM mass. The average contribution is lower than measured at other sites in the S.E.
USA, including both rural LRK, (Budisulistiorini et al., 2015; Hu et al., 2015) and urban Atlanta,
GA (Budisulistiorini et al., 2013). The contribution of SOA tracers to OM in the current study was
estimated on the basis of offline analysis of filters, while tracer estimates in the two earlier studies
was based on online ACSM/AMS measurements. The low isoprene SOA/OM ratio is consistent
with the low WSOC/OC reported in section 3.1, suggesting an increased contribution of primary
OA or secondary OM to the total OM at BHM. However, it should be noted that total IEPOX-
derived SOA mass at BHM may actually be closer to ~14% since recent measurements by the
Aerodyne ACSM at LRK indicated that tracers could only account for ~50% of the total IEPOX-
derived SOA mass resolved by the ACSM (Budisulistiorini et al., 2015). Unfortunately, an
Aerodyne ACSM or AMS was not available at the BHM site, precluding confirmation of that
IEPOX-derived SOA mass at BHM might account for 14% (on average) of the total OM mass.
Levoglucosan, a biomass-burning tracer, averaged 1% of total OM with spikes up to 8%, the same
level measured for 2-methylthreitol and (E)-2-methylbut-3-ene-1,2,4-triol (Table 3). The ratio of
average levoglucosan at BHM relative to CTR was 5.4 suggesting significantly more biomass
burning impacting the BHM site.





IEPOX- and MAE/HMML-derived SOA tracers accounted for 18% and 0.4% of the
WSOC mass, respectively (Figure S3b), lower than the respective contributions of 24% and 0.7%
measured at LRK (Budisulistiorini et al., 2015).
Figure S5 shows no diurnal variation for the average day and night concentrations of
isoprene-derived SOA tracers. Thus cooler nighttime temperatures also do not appear to enhance
gas-to-particle partitioning at the BHM site. Figures S6 and S7 show the variation of isoprene-
derived SOA tracers during intensive sampling periods. The highest concentrations were usually
observed in samples collected from 4 pm – 7 pm, local time; however, no statistical significance
were observed between intensive periods. This observation illustrates the importance of the higher
time-resolution of the tracer data during intensive sampling periods over course of the campaign
(Table S2-S6). An additional consequence of the intensive sampling periods was resolution of a
significant correlation between isoprene SOA tracers and $O_3$ to be discussed in more detail in
section 3.3.2.
**3.3 Influence of anthropogenic emissions on isoprene-derived SOA**
**3.3.1 Effects of reactive nitrogen-containing species**
During the campaign, no isoprene-derived SOA tracers, including MAE/HMML-derived
OS and 2-MG, correlated with $NO_x$ or $NO_y$ ($r^2 = 0$, n = 120). This is inconsistent with the current
understanding of SOA formation from isoprene oxidation pathways under high-$NO_x$ conditions,
which proceeds through uptake of MAE (Lin et al., 2013a), and, as recently suggested, HMML
(Nguyen et al., 2015), to yield 2-MG and its OS derivative. Plume age, as a ratio of $NO_x:NO_y$,, in
this study was highly correlated with $O_3$ ($r^2 = 0.79$, n = 120). This correlation might be explained
by the photolysis of $NO_2$, which is abundant due to traffic at the urban ground site, resulting in





formation of tropospheric $O_3$. A negative correlation coefficient ($r = -0.47$, $n = 120$) between
plume age and 2-MG abundance was found, suggesting that formation of some 2-MG may be
associated with ageing of air masses.

A previous study supported a major role for $NO_3$ in the nighttime chemistry of isoprene

(Ng et al., 2008). Correlation of IEPOX- and MAE/HMML-derived SOA with nighttime $NO_2$, $O_3$,
and $P[NO_3]$ were examined in this study (Figures 3 and 4). As shown in Figure 3f, a moderate
correlation between MAE/HMML-derived SOA and nighttime $P[NO_3]$ ($r^2 = 0.57$, $n = 40$) was
observed. The regression analysis revealed a significant correlation at the 95% confidence interval
(*p-value* < 0.05) (Table S7). This finding suggests that some MAE/HMML-derived SOA may form
locally from the reaction of isoprene with $NO_3$ radical at night. A field study reported a peak
isoprene mixing ratio in early evening (Starn et al., 1998) as the PBL height decreases at night.  As
a result, lowering PBL heights could concentrate the remaining isoprene, $NO_2$, and $O_3$ that can
continue to react during the course of the evening. 2-MG formation has been reported to be $NO_2$-
dependent via the formation and further oxidation of MPAN (Surratt et al., 2006; Chan et al.,
2010). Hence, decreasing PBL may be related to nighttime MAE/HMML-derived SOA formation
through isoprene oxidation by both $P[NO_3]$ and $NO_2$.

Although $P[NO_3]$ depends on both $NO_2$ and $O_3$ levels, $O_3$ correlates moderately with

MAE/HMML-derived SOA tracers during day ($r^2 = 0.48$, $n = 75$), but not at night ($r^2 = 0.08$, $n = $
45). The effect of $O_3$ on isoprene-derived SOA formation during daytime will be discussed further
in section 3.3.2. $NO_2$ levels correlate only weakly with MAE/HMML-derived SOA tracers, ($r^2 = $
0.26, $n = 45$) indicating that $NO_2$ levels alone do not explain the moderate correlation of $P[NO_3]$
with these tracers.  To our knowledge, correlation of $P[NO_3]$ with high-$NO_x$ SOA tracers has not





been observed in previous field studies., indicating that further work is needed to examine the
potential role of nighttime $NO_3$ radicals in forming these SOA tracers.

As shown in Figure 4f, IEPOX-derived SOA was weakly correlated ($r^2 = 0.26$, n = 40) with

nighttime $P[NO_3]$. The correlation appears to be driven by the data at the low end of the scale and
could therefore be misleading. However, Schwantes et al. (2015) demonstrated that $NO_3$-initaited
oxidation of isoprene yields isoprene nitrooxy hydroperoxides (INEs) through nighttime reaction:
$RO_2 + HO_2$, which on further oxidation yielded isoprene nitrooxy hydroxyepoxides (INHEs). The
INHEs undergo reactive uptake onto acidic sulfate aerosol to yield SOA constituents similar to
those of IEPOX-derived SOA. The present study raises the possibility that a fraction of IEPOX-
derived SOA comes from $NO_3$-initiated oxidation of isoprene at night. The work of Ng et al.
(2008) does not explain the weak association we observe here between IEPOX-derived SOA
tracers and $P[NO_3]$ as a consequence of the reactions $RO_2 + RO_2$ and $RO_2 + NO_3$ reactions
dominating in those experiments. It is now thought that $RO_2 + HO_2$ should dominate in field studies
(Schwantes et al., 2015; Paulot et al., 2009).

### 429    3.3.2 Effect of $O_3$

During the daytime, $O_3$ was moderately correlated ($r^2 = 0.48$, n = 75) with total

MAE/HMML-derived SOA (Figure 3b). This correlation was stronger ($r^2 = 0.72$, n = 30, *p-value*
< 0.05, Table S7) when filters taken during regular daytime sampling periods are considered,
suggesting that formation of MACR (a precursor to MAE and HMML) (Lin et al., 2013b; Nguyen
et al., 2015) was enhanced by oxidation of isoprene by $O_3$ (Kamens et al., 1982). $O_3$ was not
correlated ($r^2 = 0.08$, n = 45) with MAE/HMML-derived SOA at night (Figure 3e). The latter
finding is consistent with the absence of photolysis to drive the production of $O_3$. However,





residual $O_3$ may play an important role at night to form MAE/HMML-derived SOA via the P[$NO_3$]
pathway discussed in section 3.3.1.
$O_3$ was not correlated ($r^2 = 0.10$, n = 75) with IEPOX-derived SOA during daytime (Figure
4b), but weakly correlated with 2-methylerythritol ($r^2 = 0.25$, n = 30) as shown in Table S2,
especially during intensive 3 sampling periods ($r^2 = 0.34$, n = 15, Table S5). An important
observation with regard to this result is that no correlation has been found between $O_3$ and 2-
methyltetrols ($r^2 < 0.01$) in previous field studies (Lin et al., 2013b; Budisulistiorini et al., 2015).
Isoprene ozonolysis yielded 2-methyltetrols in chamber studies in the presence of acidified sulfate
aerosol (Riva et al., 2015) but $C_5$-alkene-triols were not formed by this pathway. The greatest
abundance of isoprene-derived SOA tracers in daytime samples was generally observed in
intensive 3 samples; however, there was no statistical significance observed between intensive
samples. The moderate correlation ($r^2 = 0.34$, n = 15, *p-value* < 0.05) between $O_3$ and the 2-
methyltetrols observed in intensive 3 samples occurred when $O_3$ reached maximum levels,
suggesting that ozonolysis of isoprene plays a role in 2-methyltetrol formation. Lack of correlation
between $O_3$ and $C_5$-alkene triols during intensive 3 sampling ($r^2 = 0.10$, n = 15) supports this
contention. A putative pathway is formation of hydroperoxides that partition to wet acidic sulfate
aerosols and react further to yield 2-methyltetrols. Additional work using authentic standards is
needed to validate this tentative hypothesis.
**3.3.3 Effect of particle $SO_4^{2-}$**
$SO_4^{2-}$ was moderately correlated with IEPOX-derived SOA ($r^2 = 0.36$, n = 117) and
MAE/HMML-derived SOA ($r^2 = 0.33$, n = 117) at the 95% confidence interval as shown in Table
S7. The strength of the correlations was consistent with studies at other sites across the
Southeastern U.S. (Budisulistiorini et al., 2013; Lin et al., 2013b; Budisulistiorini et al., 2015; Xu





et al., 2015). Aerosol surface area provided by acidic $SO_4^{2-}$ has been demonstrated to control the
uptake of isoprene-derived epoxides (Lin et al., 2012; Gaston et al., 2014; Nguyen et al., 2014;
Riedel et al., 2015).

Furthermore, $SO_4^{2-}$ is proposed to enhance IEPOX-derived SOA formation by providing

particle water ($H_2O_{ptcl}$) required for IEPOX uptake (Xu et al., 2015). Aerosol $SO_4^{2-}$ also promotes
acid-catalyzed ring-opening reactions of IEPOX by $H^+$, proton donors such as $NH_4^+$, and
nucleophiles (e.g., $H_2O$, $SO_4^{2-}$, or $NO_3^-$) (Surratt et al., 2010; Nguyen et al., 2014). Since $SO_4^{2-}$
tends to drive both particle water and acidity (Fountoukis and Nenes, 2007), the extent to which
each influences isoprene SOA formation during field studies remains unclear. Multivariate linear
regression analysis on SOAS data from the CTR site and the SCAPE dataset revealed a statistically
significant positive linear relationship between $SO_4^{2-}$ and the isoprene (IEPOX)-OA factor
resolved by positive matrix factorization (PMF). On the basis of this analysis the abundance of
$SO_4^{2-}$ was concluded to control directly the isoprene SOA formation over broad areas of the
Southeastern U.S. (Xu et al., 2015), consistent with previous reports (Lin et al., 2013;
Budisulistiorini et al., 2013; Budisulistiorini et al., 2015).  Another potential pathway for $SO_4^{2-}$
levels to enhance isoprene SOA formation is through salting-in effects; however, systematic
investigations of this effect are lacking and further studies are warranted (Xu et al., 2015).
**3.3.4 Effect of aerosol acidity**

The aerosol at BHM was acidic throughout the SOAS campaign (pH range 1.60 – 1.94,

average 1.76) in accord with a study by Guo et. al. (2014) that found aerosol pH ranging from
0 – 2 throughout the southeastern U.S. However, no correlation of pH with isoprene SOA
formation was observed at BHM, also consistent with previous findings using the thermodynamic





models to estimate aerosol acidity in many field sites across the southeastern U.S. region, including
Yorkville, GA (YRK) (Lin et al., 2013b), Jefferson Street, GA (JST) (Budisulistiorini et al., 2013),
and LRK (Budisulistiorini et al., 2015). However, it is important to point out that the lack of
correlation between SOA tracers and acidity may stem from the small variations in aerosol acidity
throughout the campaign. Gaston et al. (2014) and Riedel et al. (2015) recently demonstrated that
an aerosol pH < 2 at atmospherically-relevant aerosol surface areas would allow reactive uptake
of IEPOX onto acidic (wet) sulfate aerosol surfaces to be competitive with other loss processes
(e.g., deposition and reaction of IEPOX with OH). In fact, it was estimated that under such
conditions IEPOX would have a lifetime of ~ 5 hr. The constant presence of acidic aerosol has
also been observed at other field sites in the southeastern U.S. (Budisulistiorini et al., 2013;
Budisulistiorini et al., 2015; Xu et al., 2015), supporting a conclusion that acidity is not the limiting
variable in forming isoprene SOA.
**3.4 Comparison among different sampling sites during 2013 SOAS campaign**

Table 5 summarizes the mean concentration and contribution of each isoprene SOA tracer

at BHM, CTR, and LRK. BHM is an industrial-residential area, LRK and CTR are rural areas,
although LRK is influenced by a diurnal upslope/downslope cycle of air from an urban locality
(Knoxville) (Tanner et al., 2005). IEPOX-derived SOA was predominant at all three sites during
the SOAS campaign, while MAE/HMML-derived SOA constituted a minor contribution. The
average ratio of 2-methyltetrols to $C_5$-alkene triols at BHM was 2.2, nearly double that of CTR
(1.3) and LRK (1.1). Although 2-methyltetrols and $C_5$-alkene triols are considered to form readily
from the acid-catalyzed reactive uptake and multiphase chemistry of IEPOX (Edney et al., 2005;
Surratt et al., 2006), Riva et al. (2015) recently demonstrated that only 2-methyltetrols can be
formed via isoprene ozonolysis in the presence of acidic sulfate aerosol. The higher levels of the



2-methyltetrols observed at the urban BHM site indicates a likely competition between the IEPOX
uptake and ozonolysis pathways. Together, these findings suggest that urban $O_3$ may play an
important role in forming the 2-methyltetrols observed at BHM. There were notable trends found
among the three sites: (1) average $C_5$-alkene triol concentrations were higher at CTR (214.1 ng m$^-$
$^3$) than at BHM (169.7 ng m$^{-3}$) and LRK (144.4 ng m$^{-3}$); (2) average isomeric 3-MeTHF-diol
concentrations were lower at CTR (0.2 ng m$^{-3}$) than the BHM (15.4 ng m$^{-3}$) or LRK (4.4 ng m$^{-3}$)
sites. Except for the 2-methyltetrols, reasons for the differences observed for the other tracers
between sites remains unclear and warrant future investigations.

## 4.    Conclusions

This study examined isoprene SOA tracers in $PM_{2.5}$ samples collected at the BHM ground
site during the 2013 SOAS campaign and revealed the complexity and potential multitude of
chemical pathways leading to isoprene SOA formation. Isoprene SOA contributed up to ~20%
(~7% on average) of total OM mass. IEPOX-derived SOA tracers were responsible for 92% of the
total quantified isoprene SOA tracer mass, with 2-methyltetrols being the major component (47%).
Differences in the relative contributions of IEPOX- and MAE/HMML-derived SOA tracers at
BHM and the rural CTR and LRK sites (Budisulistiorini et al., 2015) during the 2013 SOAS
campaign, support suggestions that anthropogenic emissions effect isoprene SOA formation. The
correlation between 2-methyltetrols and $O_3$ at BHM is in accord with work by Riva et al. (2015),
demonstrating a potential role of $O_3$ in generating isoprene-derived SOA in addition to the
currently accepted IEPOX multiphase pathway.
At BHM, the statistical correlation of particulate $SO_4^{2-}$ with IEPOX- ($r^2 = 0.36$, n = 117, $p$
$< 0.05$) and MAE-derived SOA tracers ($r^2 = 0.33$, n = 117, $p < 0.05$) suggests that $SO_4^{2-}$ plays a



role in isoprene SOA formation. Although none of isoprene-derived SOA tracers correlated with
gas-phase $NO_x$ and $NO_y$, MAE/HMML-derived SOA tracers correlated with nighttime $P[NO_3]$ ($r^2$
$= 0.57$, $n = 400$),   indicating that $NO_3$ may affect local MAE/HMML-derived SOA formation.
Nighttime $P[NO_3]$ was weakly correlated ($r^2 = 0.26$, $n = 40$) with IEPOX-derived SOA tracers,
lending some support to recent work by Schwantes et al. (2015) showing that isoprene + $NO_3$
yields INHEs that can by undergo reactive uptake to yield IEPOX tracers and contribute to IEPOX-
derived SOA tracer loadings. In addition, nighttime 2-methyltetrol levels in the urban atmosphere
deviate from the conventional understanding of isoprene SOA formation in terms of segregated
$NO_x$ dependent regimes. The correlation of daytime $O_3$ with MAE/HMML-derived SOA and with
2-methyltetrols offers a new insight into influences on isoprene SOA formation. Notably, $O_3$ has
not been reported to correlate with isoprene-derived SOA tracers in previous field studies (Lin et
al., 2013b; Budisulistiorini et al., 2015). In this study, the strong correlation ($r^2 = 0.72$, $n = 30$) at
the 95% confidence interval of $O_3$ with MAE/HMML-derived SOA tracers during the regular
daytime sampling schedule indicates that $O_3$ likely oxidizes some isoprene to MACR as precursor
of 2-MG at BHM. The weak correlation ($r^2 = 0.16$, $n = 75$) between $O_3$ and 2-methyltetrols early
in the day as well as the better correlation ($r^2 = 0.34$, $n = 15$) later in the day (intensive 3, 4-7 PM
local time) are consistent with recent laboratory studies demonstrating that 2-methyltetrols can be
formed via isoprene ozonolysis in the presence of acidified sulfate aerosol (Riva et al., 2015).

Although urban $O_3$ and nighttime $P[NO_3]$ may have a role in local formation of

MAE/HMML- and IEPOX-derived SOA tracers at BHM, this does not appear to explain the
majority of the SOA tracers, since no significant day-night variation of the entire group of tracers
was observed during the campaign. The majority of IEPOX-derived SOA was likely formed when
isoprene SOA precursors (IEPOX) were generated upwind and transported to the BHM site. Wind



directions during the campaign are consistent with long-range transport of isoprene SOA
precursors from southwest of the site, which is covered by forested areas. The absence of a
correlation of aerosol acidity with MAE/HMML- and IEPOX-derived SOA tracers indicates that
acidity is not the limiting variable that controls formation of these compounds. However, the lack
of correlation between SOA tracers and acidity may stem from nearly invariant aerosol acidity
throughout the campaign. Hence, despite laboratory studies demonstrating that aerosol acidity can
enhance isoprene SOA formation (Surratt et al., 2007; Surratt et al., 2010; Lin et al., 2012), the
effect may not be significant in the southeastern U.S. during the summer months due to the constant
acidity of aerosols. Future work should examine how well current models can predict the isoprene
SOA levels observed during this study, especially since urban emissions are directly present.
Furthermore, explicit models are now available to predict the isoprene SOA tracers measured here
(McNeill et al., 2012; Pye et al., 2013), which will allow the modeling community to test the
current parameterizations that are used to capture the enhancing effect of anthropogenic pollutants
on isoprene-derived SOA formation. In addition, the significant correlations of isoprene-derived
SOA tracers with $P[NO_3]$ observed during this study indicate a need to better understand nighttime
chemistry of isoprene. Lastly, although $O_3$ appears to have an enhancing effect on isoprene-
derived SOA tracers, the intermediates are unknown. Hydroperoxides suggested by Riva et al.
(2015) may be key, but chamber experiments with authentic precursors are needed to test this
hypothesis.



**Acknowledgements**
This work was funded by the U.S. Environmental Protection Agency (EPA) through grant number
835404. The contents of this publication are solely the responsibility of the authors and do not
necessarily represent the official views of the U.S. EPA. Further, the U.S. EPA does not endorse
the purchase of any commercial products or services mentioned in the publication. The authors
would also like to thank the Electric Power Research Institute (EPRI) for their support. This study
was supported in part by the National Oceanic and Atmospheric Administration (NOAA) Climate
Program Office's AC4 program, award number NA13OAR4310064.  The authors thank the
Camille and Henry Dreyfus Postdoctoral Fellowship Program in Environmental Chemistry for
their financial support. The authors thank Louisa Emmons and Christoph Knote for their assistance
with chemical forecasts made available during the SOAS campaign. We would like to thank
Annmarie Carlton, Joost deGouw, Jose Jimenez, and Allen Goldstein for helping to organize the
SOAS campaign and coordinating communication between ground sites. UPLC/ESI-HR-Q-
TOFMS analyses were conducted in the UNC-CH Biomarker Mass Facility located within the
Department of Environmental Sciences and Engineering, which is a part of the UNC-CH Center
for Environmental Health and Susceptibility supported by National Institute for Environmental
Health Sciences (NIEHS), grant number 5P20-ES10126. WSOC measurements at the University
of Iowa were supported through EPA STAR grant 8354101. The authors thank Theran Riedel for
useful discussions. We also thank SCG Chemicals Co., Ltd., Siam Cement Group, Thailand, for
the full support for W. Rattanavaraha attending UNC, Chapel Hill.




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



**Table 1.** Sampling schedule during SOAS at the BHM ground site.

| No. of samples/ day | Sampling schedule | Dates |
|---|---|---|
| 2 (regular) | Day: 8 am – 7 pm<br>Night: 8 pm – 7 am next day | June 1 – June 9<br>June 13,<br>June 17 – June 28,<br>July 2- July 9,<br>July 15 |
| 4 (intensive) | Intensive 1: 8 am – 12 pm,<br>Intensive 2: 1 pm – 3 pm,<br>Intensive 3: 4 pm – 7 pm,<br>Intensive 4: 8 pm – 7 am next day | June 10 – June 12,<br>June 14 – June 16,<br>June 29 – June 30,<br>July 1,<br>July 9 – July 14 |


**Table 2.** Summary of collocated measurements of meteorological variables, gaseous species, and
$PM_{2.5}$ constituents.

| Category | Condition | Average | SD | Minimum | Maximum |
|---|---|---|---|---|---|
| Meteorology | Rainfall (in) | 0.1 | 0.2 | 0.0 | 1.4 |
| | Temp (°C) | 26.4 | 3.0 | 20.5 | 32.7 |
| | RH (%) | 71.5 | 15.0 | 36.9 | 96.1 |
| | BP (mbar) | 994.2 | 3.9 | 984.2 | 1002.4 |
| | SR (W m$^{-2}$) | 303.7 | 274.5 | 7.0 | 885.0 |
| Trace gas (ppbv) | $O_3$ | 31.1 | 14.8 | 8.3 | 62.2 |
| | CO | 208.7 | 72.0 | 99.6 | 422.9 |
| | $SO_2$ | 0.9 | 0.8 | 0.1 | 3.7 |
| | NO | 1.3 | 1.2 | 0.1 | 7.0 |
| | $NO_2$ | 6.6 | 5.1 | 1.0 | 22.7 |
| | $NO_x$ | 7.8 | 6.0 | 1.3 | 29.7 |
| | $NO_y$ | 9.1 | 5.8 | 2.2 | 30.4 |
| | $HNO_3$ | 0.3 | 0.2 | 0.1 | 1.0 |
| | $NH_3$ | 1.9 | 0.8 | 0.7 | 4.0 |
| $PM_{2.5}$ ($\mu$g m$^{-3}$) | OC | 7.2 | 3.2 | 1.4 | 14.9 |
| | EC | 0.6 | 0.5 | 0.1 | 2.7 |
| | WSOC | 4.0 | 1.8 | 0.5 | 7.5 |
| | $SO_4^{2-}$ | 2.0 | 0.9 | 0.4 | 4.9 |
| | $NO_3^-$ | 0.1 | 0.1 | 0.0 | 0.8 |
| | $NH_4^+$ | 0.7 | 0.3 | 0.2 | 1.2 |
| | Aerosol pH | 1.8 | 0.1 | 1.6 | 1.9 |



**Table 3.** Summary of isoprene-derived SOA tracers measured by GC/EI-MS and UPLC/ESI-HR-QTOFMS

| SOA tracers | m/z | Frequency of detection (%)[a] | Max concentration (ng/m³) | Mean concentration (ng/m³) | Isoprene SOA Mass fraction (%)[b] | % of total OM[c] |
|---|---|---|---|---|---|---|
| **Measured by GC/EI-MS** | | | | | | |
| 2-methylerythritol[d] | 219 | 99.2 | 1048.9 | 269.0 | 33.8 | 2.7 |
| 2-methylthreitol[d] | 219 | 100.0 | 388.9 | 107.3 | 13.5 | 1.1 |
| (E)-2-methylbut-3-ene-1,2,4-triol[e] | 231 | 96.7 | 878.9 | 112.7 | 14.2 | 1.1 |
| (Z)-2-methylbut-3-ene-1,2,4-triol[e] | 231 | 95.8 | 287.8 | 38.9 | 4.9 | 0.4 |
| 2-methylbut-3-ene-1,2,3-triol[e] | 231 | 94.2 | 503.3 | 28.9 | 3.6 | 0.3 |
| 2-methylglyceric acid[d] | 219 | 93.3 | 35.0 | 10.8 | 1.4 | 0.1 |
| *cis*-3-MeTHF-3,4-diol[d] | 262 | 22.5 | 98.9 | 6.9 | 0.9 | 0.1 |
| *trans*-3-MeTHF-3,4-diol[d] | 262 | 10.0 | 137.6 | 8.6 | 1.1 | 0.1 |
| IEPOX-derived dimer[e] | 333 | 10.0 | 2.2 | 0.0 | 0.0 | 0.0 |
| Levoglucosan[d] | 204 | 100.0 | 922.6 | 98.7 | - | 1.0 |
| **Measured by UPLC/ESI-HR-QTOFMS** | | | | | | |
| IEPOX-derived OSs | | | | | | |
| $C_5H_{11}O_7S^{-}$ [d] | 215 | 100.0 | 864.9 | 164.5 | 20.7 | 1.6 |
| $C_{10}H_{21}O_{10}S^{-}$ [f] | 333 | 1.7 | 0.3 | 0.0 | 0.0 | 0.0 |
| MAE-derived OS[d] | | | | | | |
| $C_4H_7O_7S^{-}$ | 199 | 100.0 | 35.7 | 7.2 | 1.9 | 0.1 |
| GA sulfate[d] | | | | | | |
| $C_2H_3O_6S^{-}$ | 155 | 100.0 | 75.2 | 26.2 | 3.3 | 0.3 |
| Methylglyoxal-derived OS[g] | | | | | | |
| $C_3H_5O_6S^{-}$ | 169 | 97.5 | 10.5 | 2.7 | 0.3 | 0.0 |
| Isoprene-derived OSs[g] | | | | | | |
| $C_5H_7O_7S^{-}$ | 211 | 97.5 | 5.2 | 1.4 | 0.2 | 0.0 |
| $C_5H_{10}NO_9S^{-}$ | 260 | 90.0 | 3.9 | 0.3 | 0.0 | 0.0 |
| $C_5H_9N_2O_{11}S^{-}$ | 305 | 5.0 | 3.3 | 2.9 | 0.4 | 0.0 |
| Hydroxyacetone-derived OS[g] | | | | | | |
| $C_2H_3O_5S^{-}$ | 139 | 30.8 | 2.6 | 0.2 | 0.0 | 0.0 |

[a] Total filters = 120
[b] Mass fraction is the contribution of each species among total known isoprene-derived SOA mass detected by GC/EI MS and UPLC/ESI-HR-QTOFMS
[c] OM/OC = 1.6
[d] OA tracers quantified by authentic standards
[e] SOA tracers quantified by 2-methyltetrols as a surrogate standard
[f] SOA tracer quantified by IEPOX-derived OS (*m/z* 215) as a surrogate standard
[g] SOA tracers quantified by propyl sulfate as a surrogate standard





**Table 4.** Overall correlation ($r^2$) of isoprene-derived SOA tracers and collocated measurements at
BHM during 2013 SOAS campaign.

| SOA tracers | CO | $O_3$ | $NO_x$ | $NO_y$ | $SO_2$ | $NH_3$ | $SO_4$ | $NO_3$ | $NH_4$ | OC | WSOC | pH |
|---|---|---|---|---|---|---|---|---|---|---|---|---|
| **MAE/HMML-derived SOA tracers** | **0.07** | **0.26** | **0.00** | **0.01** | **0.06** | **0.11** | **0.33** | **0.01** | **0.18** | **0.47** | **0.20** | **0.00** |
| 2-methylglyceric acid | 0.01 | 0.26 | 0.01 | 0.00 | 0.01 | 0.07 | 0.10 | 0.00 | 0.06 | 0.19 | 0.02 | 0.00 |
| MAE-derived OS | 0.10 | 0.14 | 0.00 | 0.02 | 0.07 | 0.09 | 0.38 | 0.01 | 0.18 | 0.32 | 0.23 | 0.01 |
| **IEPOX-derived SOA tracers** | **0.04** | **0.05** | **0.00** | **0.01** | **0.05** | **0.01** | **0.36** | **0.00** | **0.21** | **0.24** | **0.12** | **0.00** |
| 2-methylerythritol | 0.00 | 0.16 | 0.03 | 0.02 | 0.01 | 0.00 | 0.30 | 0.02 | 0.18 | 0.18 | 0.19 | 0.00 |
| 2-methylthreitol | 0.00 | 0.13 | 0.02 | 0.03 | 0.02 | 0.00 | 0.20 | 0.01 | 0.16 | 0.17 | 0.15 | 0.00 |
| (E)-2-methylbut-3-ene-1,2,4-triol | 0.07 | 0.00 | 0.02 | 0.01 | 0.07 | 0.00 | 0.15 | 0.00 | 0.19 | 0.11 | 0.04 | 0.00 |
| (Z)-2-methylbut-3-ene-1,2,4-triol | 0.04 | 0.00 | 0.00 | 0.00 | 0.06 | 0.00 | 0.28 | 0.00 | 0.20 | 0.04 | 0.00 | 0.00 |
| 2-methylbut-3-ene-1,2,3-triol | 0.02 | 0.00 | 0.03 | 0.00 | 0.00 | 0.02 | 0.32 | 0.01 | 0.03 | 0.17 | 0.04 | 0.00 |
| IEPOX-derived OS | 0.02 | 0.14 | 0.03 | 0.00 | 0.00 | 0.00 | 0.27 | 0.00 | 0.16 | 0.29 | 0.29 | 0.00 |
| IEPOX dimer | 0.00 | 0.00 | 0.00 | 0.00 | 0.00 | 0.00 | 0.00 | 0.00 | 0.00 | 0.00 | 0.00 | 0.00 |
| **Other isoprene SOA tracers** GA sulfate | | | | | | | | | | | | |
| $C_2H_3O_6S^-$ | 0.30 | 0.23 | 0.01 | 0.00 | 0.08 | 0.09 | 0.27 | 0.00 | 0.19 | 0.38 | 0.18 | 0.00 |
| Methylglyoxal-derived OS | | | | | | | | | | | | |
| $C_3H_5O_6S^-$ | 0.14 | 0.04 | 0.02 | 0.03 | 0.03 | 0.07 | 0.31 | 0.02 | 0.25 | 0.21 | 0.24 | 0.00 |
| Isoprene-derived OSs | | | | | | | | | | | | |
| $C_5H_7O_7S^-$ | 0.01 | 0.23 | 0.03 | 0.01 | 0.00 | 0.02 | 0.21 | 0.00 | 0.16 | 0.31 | 013 | 0.00 |
| $C_5H_{10}NO_9S^-$ | 0.17 | 0.00 | 0.12 | 0.14 | 0.10 | 0.14 | 0.31 | 0.16 | 0.23 | 0.20 | 0.07 | 0.00 |
| $C_5H_9N_2O_{11}S^-$ * | 0.32 | 0.71 | 0.66 | 0.58 | 0.42 | 0.02 | 0.68 | 0.50 | 0.42 | 0.00 | 0.50 | 0.00 |
| Hydroxyacetone-derived OS | | | | | | | | | | | | |
| $C_2H_3O_5S^-$ | 0.02 | 0.10 | 0.08 | 0.07 | 0.05 | 0.00 | 0.00 | 0.03 | 0.00 | 0.01 | 0.01 | 0.00 |
| **Other tracer** | | | | | | | | | | | | |
| Levoglucosan | 0.00 | 0.09 | 0.02 | 0.01 | 0.02 | 0.00 | 0.00 | 0.02 | 0.00 | 0.08 | 0.04 | 0.01 |

* Found only in 6 of 120 filters
The correlations in this table are positive.










**Table 5.** Summary of isoprene-derived SOA tracers from the three SOAS ground sites: BHM,
CTR, and LRK.

| SOA tracers | Urban | | Rural | | | |
|---|---|---|---|---|---|---|
| | BHM | | CTR | | LRK | |
| | Mean (ng m⁻³) | Average amount detected tracers (%) | Mean (ng m⁻³) | Average amount detected tracers (%) | Mean (ng m⁻³) | Average amount detected tracers (%) |
| **MAE/HMML derived SOA** | | | | | | |
| MAE/HMML-derived OS | 7.2 | 1.1 | 10.2 | 1.3 | 8.2 | 1.8 |
| 2-methylglyceric acid | 10.4 | 1.7 | 5.1 | 0.7 | 7.5 | 1.6 |
| | | | | | | |
| **IEPOX derived SOA** | | | | | | |
| IEPOX-derived OS | 164.5 | 24.3 | 207.1 | 26.8 | 139.2 | 30.3 |
| IEPOX-derived dimer OS | 0.04 | 0.00 | 0.7 | 0.1 | 1.1 | 0.2 |
| 2-methylerythritol | 266.7 | 37.9 | 204.8 | 26.5 | 120.7 | 26.3 |
| 2-methylthreitol | 107.3 | 15.8 | 73.7 | 9.5 | 42.4 | 9.2 |
| (E)-2-methylbut-3-ene-1,2,4-triol | 109.0 | 12.3 | 137.3 | 17.8 | 98.8 | 21.5 |
| (Z)-2-methylbut-3-ene-1,2,4-triol | 37.3 | 4.1 | 50.7 | 6.6 | 29.1 | 6.1 |
| 2-methylbut-3-ene-1,2,3-triol | 23.4 | 2.5 | 26.1 | 3.4 | 16.5 | 3.6 |
| trans-3-MeTHF-3,4-diol | 8.6 | 1.0 | 0.0 | 0.0 | 2.7 | 0.6 |
| cis-3-MeTHF-3,4-diol | 6.8 | 1.0 | 0.2 | 0.0 | 1.7 | 0.4 |
















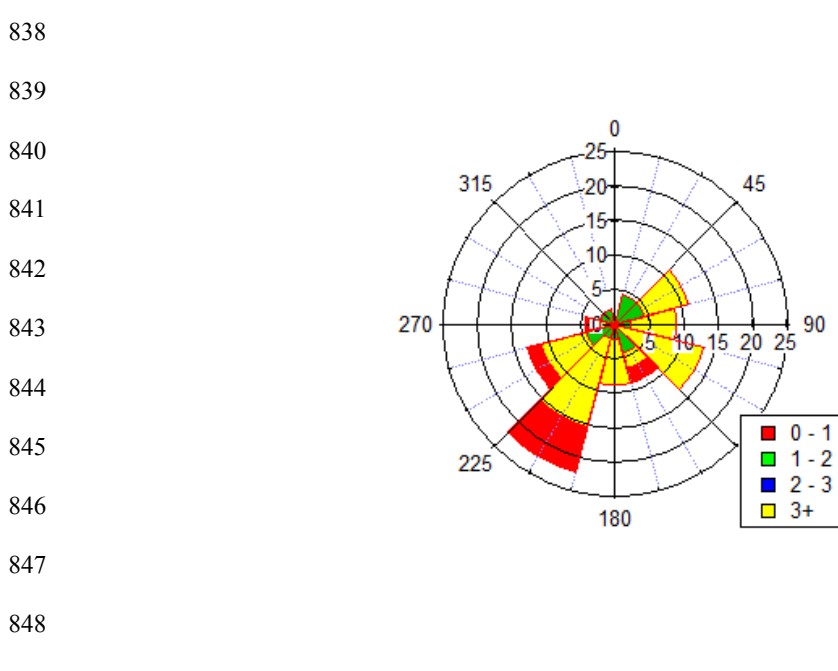

**Figure 1.** Wind rose illustrating wind direction during the campaign at the BHM site. Bars indicate direction of incoming wind, with 0 degrees set to geographic north. Length of bar size indicates frequency with color segments indicating the wind speed in m s$^{-1}$.



**Figure 2.** Time series of (a) meteorological data, (b) trace gases, (c) PM$_{2.5}$ constituents, (d) MAE/HMML-derived SOA tracers and (e) IEPOX-derived SOA tracers during the 2013 SOAS campaign at the BHM site.


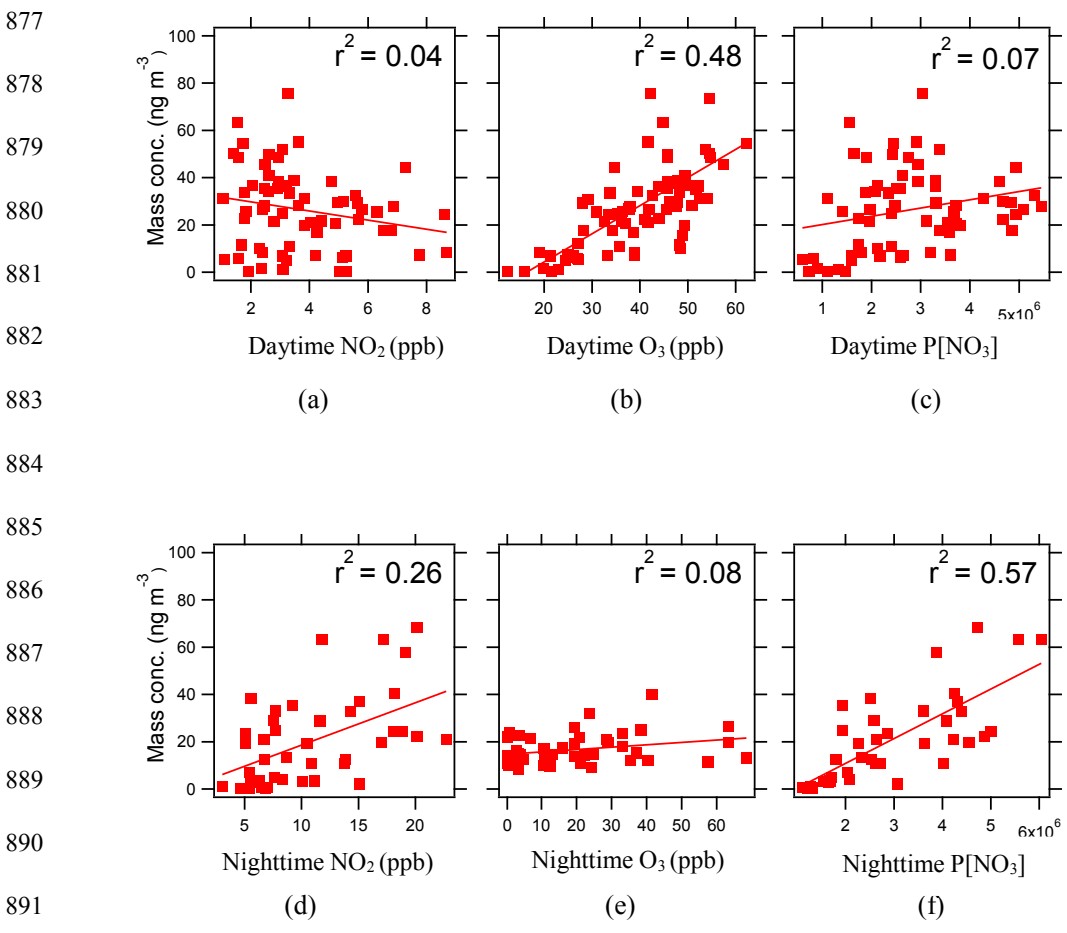

(a)              (b)              (c)

(d)              (e)              (f)

**Figure 3.** Correlation of MAE-derived SOA tracers with (a) daytime $NO_2$, (b) daytime $O_3$, (c)
daytime $P[NO_3]$, (d) nighttime $NO_2$, (e) nighttime $O_3$, and (f) nighttime $P[NO_3]$. Nighttime $P[NO_3]$
correlation suggests that $NO_3$ radical chemistry could explain some fraction of the MAE/HMML-
derived SOA tracer concentrations.

















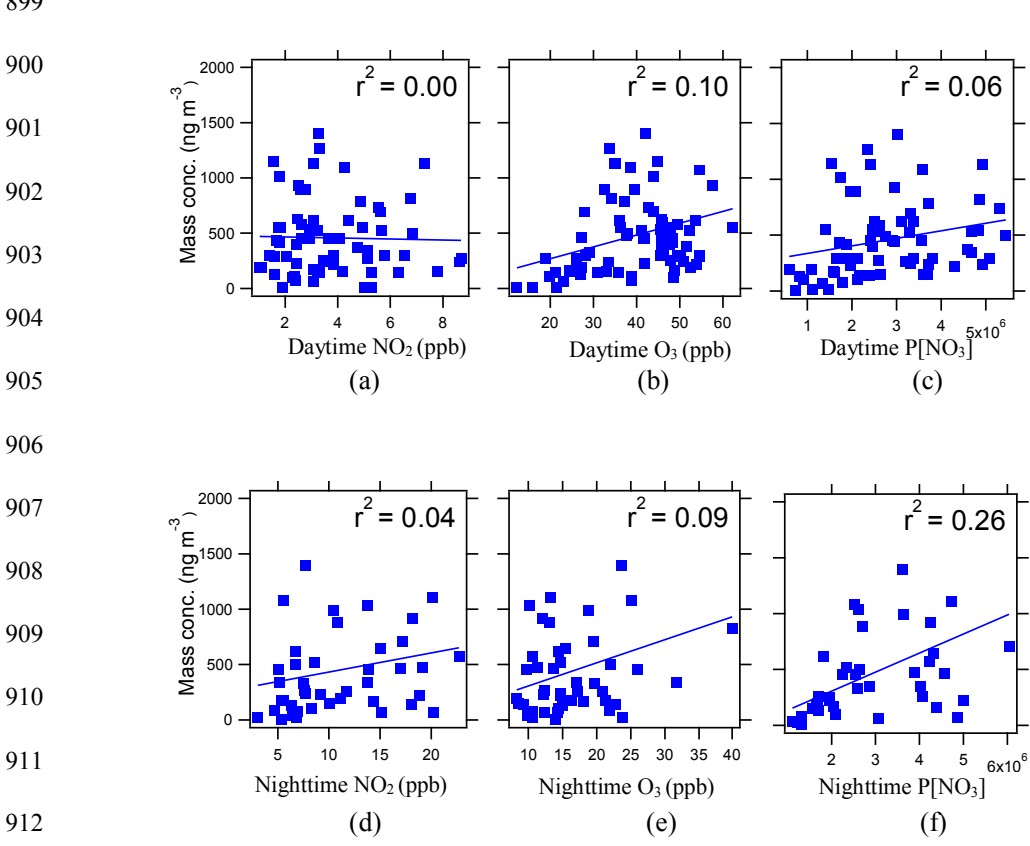

**Figure 4.** Correlation of IEPOX-derived SOA tracers with (a) daytime $NO_2$, (b) daytime $O_3$ , (c) daytime $P[NO_3]$, (d) nighttime $NO_2$, (e) nighttime $O_3$, and (f) nighttime $P[NO_3]$. Nighttime $P[NO_3]$ correlation suggests that $NO_3$ radical chemistry could explain some fraction of the IEPOX-derived SOA tracer concentrations.