# Peer review of "Assessing the impact of anthropogenic pollution on isoprene-derived secondary organic"

_Atmospheric Chemistry and Physics, 2015_

## Referee Comment (RC1) · Anonymous Referee #4 · 15 Feb 2016

This paper presents novel data from solvent-extracted filter-collected aerosol in the southeastern United States during the SOAS campaign in summer 2013, which has been analyzed to understand the distribution of isoprene oxidation products, as well as correlated against other measurements to elucidate formation mechanisms of these species. This is a good contribution to understanding of anthropogenic effects on SOA formation from isoprene, and I recommend publication after minor corrections & consideration of a few questions.
[Figure]

Questions:

1) On p. 16 you describe the slightly lower contribution of the low-NOx pathway tracers at your BHM urban site – 93% compared to 97-98% at the more rural sites. Do I understand this contribution analysis correctly to imply that at all 3 sites the overwhelming majority of isoprene SOA tracers are from the "low-NOx" pathway? Given that you site is urban, does this suggest that a rethink of the "high-NOx" / "low-NOx" split of these tracers is in order? Or, how do you understand the fact that in an urban center with 3-15 ppb NOx, only 7% of the isoprene SOA tracers appear to be "high-NOx" products?

2) The lack of diurnal variation between avg daytime and nighttime concentrations of isop-SOA tracers is interesting. Do you think this is mainly because they are long-lived and formed upwind? Or do you think there might be some offsetting daytime higher source strength and nighttime temperature-driven higher particle partitioning? Maybe add a bit of discussion of this around line 374. As I mention later, I also think the diurnal cycle/day-night comparison supplemental figures should go in the main paper.

3) Couldn't the NOx/NOy plume age correlation with O3 you mention at the beginning of 3.3.1 be just be a consequence of the relative diurnal variations you mentioned previously in NOx and O3? Thus, plume age could be actually not changing much. . . suggest thinking about this in your discussion. Related question pertaining to the negative correlation of plume age and 2-MG mentioned at the top of page 19: do you see a typical diurnal cycle of "plume age", or is the variation mostly in the day to day differences? (also related: are we looking at intensives data here or just day/night samples?) I'm wondering if this could just be saying that 2-MG has a pretty consistent diurnal cycle, with a peak in the afternoon after NOx has decreased.

4) Towards the middle and bottom of p. 19 you are talking about both NO2 and NO3 enhanced MAE/HMML derived SOA formation. You seem to be assuming that these might have similar structures – my first question: is there a known mechanism for MAE/HMML from NO3 + isoprene? Because usually NO3 initiated chemistry retains

the NO3 group, I would expect it to make different products than these. Further down in that paragraph that goes onto the next page: I don't think it's at all obvious that high-NOx SOA tracers would be the same as NO3 chemistry tracers – the nitrate group is at a different position in the molecule when formed via isopRO2+NO vs. NO3+isoprene chemistry.

Minor suggestions/edits:

1) line 52: "indicates that" => "is consistent with the observation that"

2) line 54: "the reports" =>"previous studies suggesting"?

3) line 61: remove "potential"?

4) lines 72-73: add mention of biogenic sources of VOCs here too

5) line 90-91: phrase "isomeric isoprene epoxydiols" is a bit confusing – maybe "multiple isomers of isoprene epoxydiols"?

6) line 111: "considerable" doesn't sound quantitative– maybe "large"?

7) line 120: "estimates" => "estimated"

8) lines 122-123: I think it hadn't yet been stated that IEPOX is necessarily formed in the particle phase – a brief explanation somewhere before this conclusive statement would be good.

9) around lines 129-130: does this addition only increase accuracy of isoprene SOA prediction, or total SOA prediction more generally?

10) lines 133-136: a little unclear – I think what you mean to say here is in order to develop feasible control strategies, not in order to understand?

11) line 145: mention here that you're talking about filter collected also in BHM (right?), not just as part of SOAS.

12) lines 146-148: you've already introduced these acronyms, so I think you can just

use the abbreviations here

13) line 188: suggest to add a bit more details here which (relevant) trace gases were measured, and that they were measured continuous as well

14) Around line 199: suggest adding a brief discussion here of the target functional group of the derivatization – what chemical conversion are you doing, and what class of compounds does it enable quantifying?

15) Around line 220-221: Are you analyzing derivatized or not in this case? It's unclear from the way you reference section 2.2.1. Also I think you mean to refer to section 2.2.2.

16) line 290: omit "$\sim$" in front of temperature

17) line 299: do you mean to again compare intensive days to regular days with the "lower"? if so, I recommend mentioned also the averages for intensive days, with parallel structure to the sentence above: "on intensive days, compared to ..., ... and ... on regular sampling days." Or, if you actually meant to compare to the concentration of O3 itself, I don't understand why.

18) line 307-308: don't you have a direct measurement of OC that you could also compare to the Budis and Hu 2015 references' values to confirm your hypothesis here?

19) Line 312: remove hyphen in "High-NOx" since it's not used as an adjective here.

20) line 313: "most likely in conjunction with rising O3 levels": what does this mean? are you suggesting the major NOx loss is to reaction with O3? I think rather you're making RONO2/ROONO2/HNO3 and also the BL height is increasing - and NOx emissions peak at rush hour, while O3 production cranks along all day driven by radiation. So, NO2 goes down while O3 goes up, but in my opinion, "in conjunction with" suggests a direct chemical connection that isn't likely the major reason they show the opposite trend

21) line 323: the referred to AMS here was at CTR, correct? Suggest you say so.

22) line 354: suggest "ranging up to"

23) line 359: "was" => "were"

24) line 360: "an increased" => "a larger"

25) line 365-366: "of that . . .. OM mass." awkward phrasing – suggest rewording.

26) line 367: start a new paragraph at "Levoglucosan. . ."?

27) line 369-370: more BB influence at the urban site! This surprises me – why do you think this would be the case? Is there any other confirmatory evidence of this? Or are there other possible sources in an urban area? I would have thought rural areas would have more BB contributions, because of regional crop burning. . ..

28) line 375: remove "also"

29) line 378: now you are talking about there BEING some diurnal variation, where the beginning of this paragraph talks about no difference day/night. I suggest reworking the text to clarify – I guess you're looking at different sets of samples, but it's confusing as written.

30) line 379: do you mean no stat. sig. DIFFERENCE between periods? And, do you mean between different times of day within the intensives, or between different 2-day intensive sampling periods? (I have the same question in some of the SI captions)

31) line 393: first report of an "r" instead of r^2 . . . makes comparisons tricky. Maybe just keep as r^2 but mention the correlation is negative?

32) line 404: concentration would only increase with lowering PBL height if isoprene continues to be emitted at night . is it?

33) line 406: if MPAN oxidation is responsible for 2-MG formation, you'd need to see the NO2 correlation, which you don't, correct?

34) line 419: "initiated" (spelling error)

35) lines 424-428: this isn't super clear : are you saying that Ng 2008 didn't see this correlation because they didn't have RO2+HO2 reactions, and you're attributing your observation of a weak correlation to those RO2+HO2 rxns and not RO2+RO2 or RO2+NO3, which Ng would have observed exclusively? Suggesting reworking the text.

36) line 452: suggest "putative" => "potential"

37) line 475: briefly explain "salting-in" chemistry

38) lines 485-486: "may stem from... campaign": add, or the fact that it was always plenty acidic and thus not at all pH-limited! (not just that it was relatively constant)

39) lines 498/499: depending on how you end up discussing this high-NOx/low-NOx idea, consider reminding the reader here of which products correspond to which NOx regime.

40) line 504: you mean specifically, without the intermediate of IEPOX, right? If so, say so.

41) line 522: "effect" => "affect"

42) lines 534-536: "in addition ... regimes." seems to be introducing some new ideas – be sure you say something about this above in the main text.

43) lines 439-545: "In this study ... (Riva et al., 2015)." I think the bulk of this text should go above in 3.3.2. with just a summary here - seems like you're presenting some new correlations here in the conclusions section.

44) lines 555-556: again, nearly invariant and ALWAYS very acidic is the key I think you're trying to present here.

45) line 560: "since urban emissions are directly present" => " in the presence of fresh urban emissions"

46) Table 1: the periods for the intensive aren't clear to me - the 4 sampling periods suggest coverage of 2 days, but these periods list 3 days - ?

47) in table 4: I assume the bold lines are aggregated tracers for MAE/HMML vs. IEPOX? Explain in the caption

48) Table 5: "average amount detected tracers" => "average fraction of detected tracers"

49) Fig. 2: Looks like NOx peaks are mostly during fires, based on CO spikes concurrent? Does this affect any of the plume age analysis?

It's quite hard to discern any day/night patterns here - maybe average day & night values, with SD bars, for some key metrics would be a good figure to include in the main body text? Also, add your plume age calc here to the time series? I'm curious how much it varies over the campaign vs. has a typical diurnal pattern.

50) In caption of Fig. 4: say something about this being a smaller fraction than Fig. 3 – because significantly weaker correlation.

SI: I would put S5-S7 in the main body of the paper. Also, in those captions, when you say there is "no significant variation was observed amongst intensive samples", do you mean to compare different date periods where you did the 4 time chunks, or do you mean between the 4 time chunks over all of the date periods where you did that finer time resolution, or both? please clarify

An SI figure with a couple key structures & corresponding acronyms would be nice (MAE, HMML, etc.)

---

## Referee Comment (RC2) · Anonymous Referee #1 · 16 Feb 2016

General comments:

The focus of this manuscript is on the relationships between isoprene SOA markers and anthropogenic parameters such as sulfate and NOx during a field campaign conducted in Birmingham (BHM), Alabama, an urban site in the southeastern USA, where regional isoprene emissions from deciduous trees and local anthropogenic emissions (SO2 and NOx) are substantial. The study reveals the complexity and potential multitude of chemical pathways leading to isoprene SOA formation. It is interesting to

learn that ozone also plays a role in forming isoprene SOA. It is evident from this field study that the IEPOX-related SOA markers contributed by far the most, while the methacrolein-related tracers were rather minor. It is also shown that acidity is not a limiting factor for isoprene SOA formation at the BHM site. Understanding the formation mechanisms of biogenic SOA, especially with regard to anthropogenic emissions, is indeed important for the development of more accurate models that are needed for PM2.5 control abatement strategies. I only have some specific issues (mostly minor), which would profit from clarification.

Specific comments:

Lines 229-230: The EIC of m/z 199 was used to quantify the MAE/HMML-derived OS. It has been quite well established that the major MAE-derived OS that is present in ambient fine aerosol is 3-sulfooxy-2-hydroxy-2-methylpropanoic acid with a terminal sulfooxy group (Gómez-González et al., 2010). There is, to my knowledge, no chemical evidence for a HMML-derived OS containing a terminal hydroxymethyl group. Therefore, it would be best to be more conservative here and write: ". . . . . . to quantify the MAE-derived OS, . . ..."

Ref.: Gómez-González, Y., Surratt, J. D., Cuyckens, F., Szmigielski, R., Vermeylen, R., Jaoui, M., Lewandowski, M., Offenberg, J. H., Kleindienst, T. E., Edney, E. O., Blockhuys, F., Van Alsenoy, C., Maenhaut, W., and Claeys, M.: Characterization of organosulfates from the photooxidation of isoprene and unsaturated fatty acids in ambient aerosol using liquid chromatography/(-) electrospray ionization mass spectrometry, J. Mass Spectrom., 43, 371–382, 2008.

Line 238: Mention is also made here of glyoxal-, methylglyoxal and hydroxyacetone-derived OS, but earlier in the introduction nothing is said about the chemical pathways leading to these products. For completeness, I suggest to briefly provide some background information about these pathways.

Lines 335-337: It is mentioned that the IEPOX-derived SOA tracers comprised 92.5%

of the total detected SOA tracer mass. To arrive at this value the 2-methyltetrols should not be counted twice, since they are also formed from IEPOX OS in the analytical GC/MS procedure by hydrolysis of the corresponding OS. How was this value of 92.5% estimated?

Line 340: See comment about MAE/HMML-OS above.

Lines 359-361: It is not clear what the authors want to say by writing: "The low isoprene SOA/OM ratio is consistent with the low WSOC/OC reported in section 3.1, suggesting an increased contribution of primary OA or secondary OM to the total OM at BHM". Some clarification should be given about the secondary OM; perhaps this should be better described as "hydrophobic secondary OM", originating from anthropogenic emissions.

Lines 386-408: Throughout this section, MAE/HMML OS is mentioned several times. See comment about MAE/HMML-OS above.

Section 3.3.2 Effect of O3: It is very interesting to learn that O3 has an effect on 2-methyltetrol formation, but not on C5-alkene triol formation, consistent with a recent study by Riva et al. (2015). What I am missing here is some brief mechanistic information about 2-methyltetrol formation from isoprene by the action of O3, so that the interested reader does not have to look up the original article. I am also curious to know whether there was an intercorrelation between the 2-methyltetrols and the C5-alkene triols; a strong intercorrelation would support that they both are formed through the IEPOX multiphase pathway.

Technical corrections:

Line 35: ...... coupled to electrospray ionization high-resolution quadrupole time-of-flight mass spectrometry ......

Line 148: ...... to measure known ........ or ........ to determine quantities of ........

Line 176: ...... and NOx were forecast by ......

Line 254: . . . . . . baked at 500 degrees C for . . . . . (space before degrees C)

Line 258: The abbreviation "PTFE" needs to be introduced here.

Line 352: . . . . . . based on recent studies . . . . .

Line 419: . . . . . NO3-initiated oxidation . . . . . .

References: titles of journal articles should not be capitalized; some still are capitalized and should be corrected.

Line 807: . . . . . . trans-3-methyl-3,4-dihydroxytetrahydrofuran, . . . . .

Figure 2 – panel (d): the trace for MAE-OS in light purple/rose color is hardly visible.

Please also note the supplement to this comment:
http://www.atmos-chem-phys-discuss.net/acp-2015-983/acp-2015-983-RC2-
supplement.pdf

---

## Author Comment (AC1) · 2 Apr 2016

This paper presents novel data from solvent-extracted filter-collected aerosol in the southeastern United States during the SOAS campaign in summer 2013, which has been analyzed to understand the distribution of isoprene oxidation products, as well as correlated against other measurements to elucidate formation mechanisms of these species. This is a good contribution to understanding of anthropogenic effects on SOA formation from isoprene, and I recommend publication after minor corrections & consideration of a few questions.

**Questions**

1) On p.16 you describe the slightly lower contribution of the low-$NO_x$ pathway tracers at your BHM urban site ~93% compared to 97-98% at the more rural sites. Do I understand this contribution analysis correctly to imply that at all 3 sites the overwhelming majority of isoprene SOA tracers are from the "low-$NO_x$" pathway? Given that you site is urban, does this suggest that are think of the "high-NOx"/"low-NOx" split of these tracers is in order? Or, how do you understand the fact that in an urban center with 3-15 ppb $NO_x$, only 7% of the isoprene SOA tracers appear to be "high-$NO_x$" products?

Yes, at all three sites the overwhelming majority of isoprene SOA tracers are from the low-$NO_x$ pathway (~93% at urban BHM and 97-98% at rural LRK and YRK). Approximately 2-3% of quantified isoprene SOA tracers appear to be "high-$NO_x$" products at LRK (Budisulistiorini et al., 2015) and YRK (Lin et al., 2013b). The MAE/HMML-derived OS and 2-MG may be formed upwind and transported to the rural sampling sites. As stated

below in our response to the reviewer comment # 2, a recent study at CTR demonstrated with the FIGAERO-CIMS that isoprene-derived SOA is effectively nonvolatile, so this material is likely long-lived in PM. This can result in it being transported to our sampling site. Since the vast majority of isoprene is emitted upwind, it is likely that the oxidation products formed outside of the city under lower NO conditions come into contact with urban aerosols (which includes the sulfate aerosol) to form this low-volatility isoprene SOA. Furthermore, At the BHM, 7% of the isoprene SOA tracers are high-$NO_x$ products (~3%), GA sulfate (~3%), methylglyoxal-derived OS (~0.3%), and other isoprene-derived OSs (~0.7%) as shown in Table 3. GA sulfate is observed as high as a likely "high-$NO_x$" product, since it could have additional sources other than isoprene such as anthropogenic VOCs (Galloway et al., 2009; Liao et al., 2015). The contribution of GA sulfate in this study was consistent with the level of GA sulfate measured by the airborne NOAA Particle Analysis Laser Mass Spectrometer (PALMS) over the continental U.S. during the Deep Convection Clouds and Chemistry Experiment and SEAC4RS (Liao et al., 2015). However, GA sulfate and methylglyoxal-derived OS can form from biogenic and anthropogenic emissions other than isoprene (Galloway et al., 2009; Liao et al., 2015). For this reason, GA sulfate and methylglyoxal-derived OS are not further discussed in this study.

2) The lack of diurnal variation between avg daytime and nighttime concentrations of isoprene-SOA tracers is interesting. Do you think this is mainly because they are long-lived and formed upwind? Or do you think there might be some offsetting daytime higher source strength and nighttime temperature-driven higher particle partitioning? Maybe add a bit of discussion of this around line 374. As I mention later, I also think the diurnal cycle/day-night comparison supplemental figures should go in the main paper.

We agree with the referee's comments. We have moved the diurnal cycle/day night comparison figures (now Figures 3-5) from supplemental information into the main text as the referee suggested.

We have also added some discussions as follows on Pages 18-19, Lines 401-414:
"*Figure 3 shows no difference for the average day and night concentration of isoprene-derived SOA tracers, suggesting that the majority of isoprene SOA tracers are potentially long-lived and formed upwind. A recent study by Lopez-Hilfiker et al. (2016) at the CTR site*

*during the 2013 SOAS demonstrated that isoprene-derived SOA was comprised of effectively nonvolatile material, which could allow for this type of SOA to be long-lived in the atmosphere. Although 2-MG and MAE-derived OS are known to form under high-$NO_x$ conditions (Lin et al., 2013a), no correlation between 2-MG and MAE-derived OS with $NO_x$ (Table 4) is observed at the BHM. This supports that isoprene SOA tracers likely formed at upwind locations and subsequently transported to the sampling site. Higher isoprene emissions during the daytime and cooler nighttime temperatures do not appear to cause any differences between daytime and nighttime isoprene-derived SOA tracer concentrations. Figures 4 and 5 show the variation of isoprene-derived SOA tracers during intensive sampling periods. The highest concentrations were usually observed in samples collected from 4 pm – 7 pm, local time; however, no statistical significance were observed between intensive periods."*

3) Couldn't the $NO_x$/$NO_y$ plume age correlation with $O_3$ you mention at the beginning of 3.3.1 be just be a consequence of the relative diurnal variations you mentioned previously in $NO_x$ and $O_3$? Thus, plume age could be actually not changing much... suggest thinking about this in your discussion. Related question pertaining to the negative correlation of plume age and 2-MG mentioned at the top of page 19: do you see a typical diurnal cycle of "plume age", or is the variation mostly in the day to day differences? (Also related: are we looking at intensives data here or just day/night samples?) I'm wondering if this could just be saying that 2-MG has a pretty consistent diurnal cycle, with a peak in the afternoon after $NO_x$ has decreased.

We agree with the referee's suggestion. Please note that we are here looking at the overall data including day, night, and intensive samples together. Only the typical diurnal cycle of "plume age" is observed for comparison.

We added some revised text in Section 3.3.1 as follows on Pages 19-20, Lines 425-433:

*"Plume age, as a ratio of $NO_x$:$NO_y$, in this study was highly correlated with $O_3$ ($r^2 = 0.79$, n = 120) which is consistent with the relative diurnal variation of $NO_x$, $NO_y$, and $O_3$ as discussed in Section 3.1. This correlation might be also explained by the photolysis of $NO_2$, which is abundant due to traffic at the urban ground site, resulting in formation of*

*tropospheric $O_3$. A negative correlation coefficient ($r^2$ = 0.22, n = 120) between plume age and 2-MG abundance was found as a consequence of relative diurnal variations. The peak of 2-MG was observed in the afternoon after $NO_x$ has decreased. This correlation leads to the hypothesis that the formation of 2-MG may be associated with ageing of air masses; however, further investigation is warranted."*

4) Towards the middle and bottom of p.19 you are talking about both $NO_2$ and $NO_3$ enhanced MAE/HMML derived SOA formation. You seem to be assuming that these might have similar structures – my first question: is there a known mechanism for MAE/HMML from $NO_3$+isoprene? Because usually $NO_3$ initiated chemistry retains the $NO_3$ group, I would expect it to make different products than these. Further down in that paragraph that goes on to the next page: I don't think it's at all obvious that high-$NO_x$ SOA tracers would be the same as $NO_3$ chemistry tracers – the nitrate group is at a different position in the molecule when formed via isop $RO_2$+NO vs. $NO_3$+isoprene chemistry.

To our knowledge, the mechanism for MAE/HMML from $NO_3$ + isoprene is still unknown. We agree with the referee's suggestion that high-$NO_x$ SOA tracers would not be the same as $NO_3$ chemistry tracers. Thus, we only reported the correlation we have observed at the site in this study and note that further work is needed to examine the potential role of nighttime $NO_3$ radicals in forming MAE/HMML-derived SOA tracers.

**Minor suggestions/edits:**

1) **Line 52:** "indicates that" => "is consistent with the observation that"

We edited the sentence at the referee suggested as follows on Page 3, Lines 51-55:

*"Lack of correlation between aerosol acidity and isoprene-derived SOA is consistent with the observation that acidity is not a limiting factor for isoprene SOA formation at the BHM site as aerosols were acidic enough to promote multiphase chemistry of isoprene-derived epoxides throughout the duration of the study."*

2) **Line 54:** "the reports" =>"previous studies suggesting"?

We edited the sentences as the referee suggested as follows on Page 3, Lines 54-55:

*"All in all, these results confirm previous studies suggesting that anthropogenic pollutants enhance isoprene-derived SOA formation."*

3) **Line 61**: remove "potential"?

We removed "potential" in front of "human health risk" as the referee suggested as follows on Page 3, Line 61-62:

*"In addition to climatic effects, PM$_{2.5}$ has been demonstrated to pose a human health risk through inhalation exposure (Pope and Dockery, 2006; Hallquist et al., 2009)."*

4) **Lines 72-73:** add mention of biogenic sources of VOCs here too

We edited the sentences as the referee suggested as follows on Page 4, Lines 72-76:

*"Processes such as natural plant growth, biomass burning and combustion also yield volatile organic compounds (VOCs), which have high vapor pressures and can undergo atmospheric oxidation to form secondary organic aerosol (SOA) through gas-to-particle phase partitioning (condensation or nucleation) with subsequent particle-phase (multiphase) chemical reactions (Grieshop et al., 2009)."*

5) **Lines 90-91:** phrase "isomeric isoprene epoxydiols" is a bit confusing-maybe "multiple isomers of isoprene epoxydiols"?

We edited the sentences as the referee suggested as follows on Page 4, Lines 90-92:

*"Under low-NO$_x$ conditions, such as in a pristine environment, multiple isomers of isoprene epoxydiols (IEPOX) have been demonstrated to be critical to the formation of isoprene SOA."*

6) **Line 111:** "considerable" doesn't sound quantitative–maybe "large"?

We edited the sentences as the referee suggested as follows on Page 6, Lines 118-119:

*"Due to the large emissions of isoprene, an SOA yield of even 1% would contribute*

*significantly to ambient SOA (Carlton et al., 2009; Henze et al., 2009)."*

7) **Line 120:** "estimates" => "estimated"

We edited the sentences as the referee suggested as follows on Pages 6, Lines 125-128:

*"The individual ground sites corroborate recent aircraft-based measurements made in the Studies of Emissions and Atmospheric Composition, Clouds, and Climate Coupling by Regional Surveys (SEAC4RS) aircraft campaign, which estimated an IEPOX-SOA contribution of 32% to OA mass in the southeastern U.S. (Hu et al., 2015)."*

8) **Lines 122-123:** I think it hadn't yet been stated that IEPOX is necessarily formed in the particle phase–a brief explanation somewhere before this conclusive statement would be good.

We thank the referee for this point. The particle-phase formation of IEPOX has been mentioned earlier on Pages 4-5, Lines 90-101, where all prior references were cited:

*"Under low-NOx conditions, such as in a pristine environment, multiple isomers of isoprene epoxydiols (IEPOX) have been demonstrated to be critical to the formation of isoprene SOA. On advection of IEPOX to an urban environment and mixing with anthropogenic emissions of acidic sulfate aerosol, SOA formation is enhanced (Surratt et al., 2006; Lin et al., 2012; Lin et al., 2013b). This pathway has been shown to yield 2-methyltetrols as major SOA constituents of ambient $PM_{2.5}$ (Claeys et al, 2004; Surratt et al., 2010; Lin et al., 2012). Further work has revealed a number of additional IEPOX-derived SOA tracers, including $C_5$-alkene triols (Wang et al., 2005; Lin et al., 2012), cis- and trans-3-methyltetrahydrofuran-3,4-diols (3-MeTHF-3,4-diols) (Lin et al., 2012; Zhang et al., 2012), IEPOX-derived organosulfates (OSs) (Lin et al., 2012), and IEPOX-derived oligomers (Lin et al., 2014). Some of the IEPOX-derived oligomers have been shown to contribute to aerosol components known as brown carbon that absorb light in the near ultraviolet (UV) and visible ranges (Lin et al., 2014)."*

9) **Around lines 129-130:** does this addition only increase accuracy of isoprene SOA prediction, or total SOA prediction more generally?

It improves both. We have revised the sentence as follows on Pages 6-7, Lines 135-140:

*"Recent work demonstrates that incorporating the specific chemistry of isoprene epoxide precursors into models increases the accuracy and amount of isoprene SOA predictions (Pye et al., 2013; Karambelas et al., 2014; McNeill., 2015), suggesting that understanding the formation mechanisms of biogenic SOA, especially with regard to the effects of anthropogenic emissions, such as NO$_x$ and SO$_2$, will be key to more accurate models."*

10) **Lines 133-136:** a little unclear– I think what you mean to say here is in order to develop feasible control strategies, not in order to understand?

By writing on Page 7, Lines 140-143: *"More accurate models are needed in order to devise cost-effective control strategies for reducing PM$_{2.5}$ levels. Since isoprene is primarily biogenic in origin, and therefore not controllable, the key to understanding the public health and environmental implications of isoprene SOA lies in resolving the effects of anthropogenic pollutants.",* we intend to understand the roles of isoprene SOA formation associated with uncontrollable biogenic emissions and controllable anthropogenic emissions, so that the control strategies will be developed in the future based on the anthropogenic emissions.

11) **Line 145:** mention here that you're talking about filter collected also in BHM (right?), not just as part of SOAS.

We introduced, in general, that primary purpose of SOAS campaign was to examine, in greater detail, the formation mechanism, composition, and properties of biogenic SOA, including the effects of anthropogenic emissions. However, this study pertains specifically to the results from the BHM site, which also served the primary purpose of the SOAS campaign and was apart of the SOAS study. We were funded by EPRI to have filters collected during SOAS as this site.

12) **Lines 146-148:** you've already introduced these acronyms, so I think you can just use the abbreviations here.

We already introduced the GC/EI-MS and UPLC/ESI-HR-QTOFMS in the abstract. Thus, as the referee suggested, we edited the sentences on Page 7, Line 151-152 as follows:

*"The results presented here focus on analysis of PM$_{2.5}$ collected on filters during the campaign by GC/EI-MS and UPLC/ESI-HR-QTOFMS."*

13) **Line 188:** suggest to add a bit more details here which (relevant) trace gases were measured, and that they were measured continuous as well.

We added some additional information as the referee suggested as follows on Page 9, Lines 191-195:

*"In addition to filter sampling of PM$_{2.5}$, SEARCH provided a suite of additional instruments at the site that measured meteorological and chemical variables, including temperature, relative humidity (RH), trace gases (i.e., CO, O$_3$, SO$_2$, NO$_x$, and NH$_3$), and continuous PM monitoring. The exact variables measured with their respective instrumentation are summarized in Table S1 of the Supplement"*

14) **Around line 199:** suggest adding a brief discussion here of the target functional group of the derivatization – what chemical conversion are you doing, and what class of compounds does it enable quantifying?

We added some information as the referee suggested as follows on Page 9, Lines 203-206:

*"The dried residues were immediately trimethylsilylated by reaction with 100 μL of BSTFA + TMCS (99:1 v/v, Supelco) and 50 μL of pyridine (anhydrous, 99.8 %, Sigma-Aldrich) at 70 °C for 1 hour. Trimethylsilyl derivatives of carbonyl and hydroxyl function groups were measurable by our GC/MS method."*

15) **Around line 220-221:** Are you analyzing derivatized or not in this case? It's unclear from the way you reference section 2.2.1. Also I think you mean to refer to section 2.2.2.

We meant to refer to Section 2.2.2 for the filter extraction procedure. We corrected the reference section as the referee suggested as follows on Page 11, Lines 233-239:

*"A 37-mm diameter circular punch from each quartz filter was extracted following the same procedure as described in Section 2.2.2 for the GC/EI-MS analysis. However, after drying,*

*the dried residues were instead reconstituted with 150 μl of a 50:50 (v/v) solvent mixture of methanol (LC-MS CHROMASOVL-grade, Sigma-Aldrich) and high-purity water (Milli-Q, 18.2 MΩ). The extracts were immediately analyzed by the UPLC/ESI-HR-QTOFMS (6520 Series, Agilent) operated in the negative ion mode. Detailed operating conditions have been described elsewhere (Riva et al., 2016). Mass spectra were acquired at a mass resolution 7000-8000."*

16) **Line 290:** omit "~" in front of temperature

We removed "~" in front of temperature as the referee suggested as follows on Page 14, Lines 310-311:

*"Temperature during this period ranged from a high of 32.6 °C to a low of 20.5 °C, with an average of 26.4 °C."*

17) **Line 299:** do you mean to again compare intensive days to regular days with the "lower"? If so, I recommend mentioned also the averages for intensive days, with parallel structure to the sentence above: "on intensive days, compared to..., ...and... on regular sampling days." Or, if you actually meant to compare to the concentration of $O_3$ itself, I don't understand why.

We only want to present the order of magnitudes of trace gases. Thus, the comparison here was generally made among different trace gases without pointing at any specific sampling time. To be clear, we removed the word "lower" in front of "were averaging 7.8" as follows on Page 15, Lines 317-320:

*"The average concentration of carbon monoxide (CO), a combustion byproduct, was 208.7 ppbv. The mean concentration of $O_3$ was significantly higher (t-test, p-value < 0.05) on intensive sampling days (37.0 ppbv) compared to regular sampling days (25.2 ppbv). Campaign average concentrations of $NO_x$, $NH_3$, and $SO_2$ were 7.8, 1.9, and 0.9 ppbv, respectively."*

18) **Line 307-308:** don't you have a direct measurement of OC that you could also compare to the Budis and Hu2015 references' values to confirm your hypothesis here?

Unfortunately, only WSOC/OC are reported in previous publications (Budisulistiorini et al.,

2015; Hu et al., 2015) used here for the comparison.

19) **Line 312:** remove hyphen in "High-$NO_x$" since it's not used as an adjective here.

We removed the hyphen in "High-$NO_x$" as the referee suggested as follows on Page 15, Lines 333-335:

"_High $NO_x$ levels were found in the early morning and decreased during the course of the day (Figure S4c), most likely due to forming $NO_x$ sinks (e.g., $RONO_2$, $ROONO_2$, and $HNO_3$) as well as possibly due to increasing planetary boundary layer (PBL) heights._"

20) **Line 313:** "most likely in conjunction with rising $O_3$ levels": what does this mean? are you suggesting the major $NO_x$ loss is to reaction with $O_3$? I think rather you're making $RONO_2/ROONO_2/HNO_3$ and also the BL height is increasing-and $NO_x$ emissions peak at rush hour, while $O_3$ production cranks along all day driven by radiation. So, $NO_2$ goes down while $O_3$ goes up, but in my opinion, "in conjunction with" suggests a direct chemical connection that isn't likely the major reason they show the opposite trend.

We have revised as the referee suggested as outlined in our response above to comment # 19.

21) **Line 323:** the referred to AMS here was at CTR, correct? Suggest you say so.

Yes, it referred to AMS at CTR. We have revised this as follows on Page 16, Lines 341-345:

"_However, the diurnal trend of isoprene levels might be similar to the data at the CTR site (Xu et al., 2015), which is only 61 miles away from BHM. Xu et al. (2015) observed the highest levels of isoprene (~ 6 ppb) at CTR in the mid-afternoon (3 pm local time) and its diurnal trend was similar to isoprene-OA measured by the Aerodyne Aerosol Mass Spectrometer (AMS) during the SOAS campaign at the CTR site._"

22) **Line 354:** suggest "ranging up to"

We agree with the referee's suggestion. We added the wording as the referee suggested on Page 17, Lines 379-381:

"_On average, isoprene-derived SOA tracers (sum of both IEPOX- and MAE/HMML-derived_

*SOA tracers) contributed ~7% (ranging up to ~ 20% at times) of the total particulate OM mass."*

23) **Line 359:** "was" => "were"

We agree with the referee's suggestion. We corrected the wording as the referee suggested on Page 17, Lines 384-385:

*"…, while tracer estimates in the two earlier studies were based on online ACSM/AMS measurements."*

24) **Line 360:** " an increased" => "a larger"

We agree with the referee's suggestion. We corrected the wording as the referee suggested on Pages 17-18, Lines 385-387:

*"The low isoprene SOA/OM ratio is consistent with the low WSOC/OC reported in Section 3.1, suggesting a larger contribution of primary OA or hydrophobic secondary OM originating from anthropogenic emissions to the total OM at BHM."*

25) **Line 365-366:** "of that....OM mass." awkward phrasing–suggest rewording.

We revised wording as the referee suggested on Page 18, Lines 391-393:

*"Unfortunately, an Aerodyne ACSM or AMS was not available at the BHM site to support the confirmation that IEPOX-derived SOA mass at BHM might account for 14% (on average) of the total OM mass."*

26) **Line 367:** start a new paragraph at "Levoglucosan..."?

We agree with the referee's suggestion. We made a new paragraph on Page 18, Lines 394-397:

*"Levoglucosan, a biomass-burning tracer, averaged 1% of total OM with spikes up to 8%, the same level measured for 2-methylthreitol and (E)-2-methylbut-3-ene-1,2,4-triol (Table 3). The ratio of average levoglucosan at BHM relative to CTR was 5.4, suggesting significantly*

*more biomass burning impacting the BHM site."*

27) **Line 369-370:** more BB influence at the urban site! This surprises me–why do you think this would be the case? Is there any other confirmatory evidence of this? Or are there other possible sources in an urban area? I would have thought rural areas would have more BB contributions, because of regional crop burning....

Although BHM is an urban site, but it's surrounded by terrestrial forests and only 61 miles away from the rural CTR sampling site. It might be possible that the BHM is affected by biomass burning around the area. The wind rose (Figure 1) illustrated that majority of the wind during the campaign came from southwest and west of the site related to terrestrial forests. An increased biomass-burning tracer at the BHM might be influenced by human activities including cooking and burning. However, investigating the sources of biomass burning is out of scope of this study.

28) **Line 375:** remove "also"

We agree with the referee's suggestion. We removed as suggested.

29) **Line 378**: now you are talking about there BEING some diurnal variation, where the beginning of this paragraph talks about no difference day/night. I suggest reworking the text to clarify–I guess you're looking at different sets of samples, but it's confusing as written.

We agree with the referee's suggestion. The revised sentences are shown as follows on Pages 18-19, Lines 401-414:

*"Figure 3 shows no difference for the average day and night concentration of isoprene-derived SOA tracers, suggesting that the majority of isoprene SOA tracers are potentially long-lived and formed upwind. A recent study by Lopez-Hilfiker et al. (2016) at the CTR site during the 2013 SOAS demonstrated that isoprene-derived SOA was comprised of effectively nonvolatile material, which could allow for this type of SOA to be long-lived in the atmosphere. Although 2-MG and MAE-derived OS are known to form under high-$NO_x$ conditions (Lin et al., 2013a), no correlation between 2-MG and MAE-derived OS with $NO_x$ (Table 4) is observed at the BHM. This supports that isoprene SOA tracers likely formed at upwind locations and subsequently transported to the sampling site. Higher isoprene*

*emissions during the daytime and cooler nighttime temperatures do not appear to cause any differences between daytime and nighttime isoprene-derived SOA tracer concentrations. Figures 4 and 5 show the variation of isoprene-derived SOA tracers during intensive sampling periods. The highest concentrations were usually observed in samples collected from 4 pm – 7 pm, local time; however, no statistical significance were observed between intensive periods."*

30) **Line 379:** do you mean no stat.sig. DIFFERENCE between periods? And, do you mean between different times of day within the intensives, or between different 2-day intensive sampling periods? (I have the same question in some of the SI captions)

We mean no significantly difference among intensive 1, 2, 3, and 4.

31) **Line 393:** first report of an "r" instead of r^2... makes comparison sticky. Maybe just keep as r^2 but mention the correlation is negative?

We agree with the referee's suggestion. The revised sentences are shown as follows on Pages 19-20, Lines 429-431:

*"A negative correlation coefficient ($r^2$ = 0.22, n = 120) between plume age and 2-MG abundance was found as a consequence of relative diurnal variations."*

32) **Line 404**: concentration would only increase with lowering PBL height if isoprene continues to be emitted at night. Is it?

No isoprene emits at night, but the remaining isoprene from daytime can carry to nighttime and will be concentrated with lowering PBL.

33) **Line 406:** if MPAN oxidation is responsible for 2-MG formation, you'd need to see the $NO_2$ correlation, which you don't, correct?

We don't see the correlation between 2-MG formation and $NO_2$ at the site, which is why we hypothesized that 2-MG might be formed upwind and transported to the site.

34) **Line 419:** "initiated" (spelling error)

We corrected a spelling error as the referee suggested.

35) **Lines 424-428**: this isn't super clear : are you saying that Ng 2008 didn't see this correlation because they didn't have $RO_2+HO_2$ reactions, and you're attributing your observation of a weak correlation to those $RO_2+HO_2$ rxns and not $RO_2+RO_2$ or $RO_2+NO_3$, which Ng would have observed exclusively? Suggesting reworking the text.

We revised the text as the referee suggested as follows Page 21, Lines 462-467:

*"The work of Ng et al. (2008), which only observed SOA as a consequence of the $RO_2 + RO_2$ and $RO_2 + NO_3$ reactions dominating the fate of the $RO_2$ radicals, does not explain the weak association between IEPOX-derived SOA tracers and P[NO3] we observe in this study. It is now thought that $RO_2 + HO_2$ should dominate the fate of $RO_2$ radicals in the atmosphere (Paulot et al., 2009; Schwantes et al., 2015)."*

36) **Line 452:** suggest "putative" => "potential"

We corrected the wording as the referee suggested.

37) **Line 475:** briefly explain "salting-in" chemistry

We added a briefly explanation of salting-in in the text as follows on Pages 23-24, Lines 521-525:

*"Another potential pathway for $SO_4^{2-}$ levels to enhance isoprene SOA formation is through salting-in effects, which the solubility of polar organic compounds would be increased in aqueous solution with increasing salt concentration (Xu et al., 2015). However, systematic investigations of this effect are lacking and further studies are warranted."*

38) **Lines 485-486:** "may stem from... campaign": add, or the fact that it was always plenty acidic and thus not at all pH-limited! (not just that it was relatively constant)

We agree with the referee's suggestion. We added the information in the text as follows on Page 24, Lines 533-535:

*"However, it is important to point out that the lack of correlation between SOA tracers and acidity may stem from the small variations in aerosol acidity and the fact that aerosols are*

*very acidic throughout the campaign."*

39) **Lines 498/499:** depending on how you end up discussing this high-NOx/low-NOx idea, consider reminding the reader here of which products correspond to which NOx regime.

We added the information the referee suggested as follows on Page 25, Lines 547-550:

*"IEPOX-derived SOA (isoprene SOA produced under low-NO$_x$ conditions) was predominant at all three sites during the SOAS campaign, while MAE/HMML-derived SOA (isoprene SOA produced under high-NO$_x$ conditions) constituted a minor contribution."*

40) **Line 504:** you mean specifically, without the intermediate of IEPOX, right? If so, say so.

We added the information the referee suggested as follows on Page 25, Lines 553-557:

*"Riva et al. (2016) recently demonstrated that only 2-methyltetrols can be formed via isoprene ozonolysis in the presence of acidic sulfate aerosol. The detailed mechanism explaining isoprene ozonolysis is still unclear, but acid-catalyzed heterogeneous reaction with organic peroxides or H$_2$O$_2$ was considered to be possible routes for 2-methyltetrol formation."*

41) **Line 522:** "effect"=>"affect"

We corrected the word as the referee suggested on Page 26, Lines 573-575:

*"Differences in the relative contributions of IEPOX- and MAE/HMML-derived SOA tracers at BHM and the rural CTR and LRK sites (Budisulistiorini et al., 2015) during the 2013 SOAS campaign, support suggestions that anthropogenic emissions affect isoprene SOA formation."*

42) **Lines 534-536:** "in addition... regimes." Seems to be introducing some new ideas –be sure you say something about this above in the main text.

We agree with the referee's suggestion. We removed the following statement because we didn't discuss about 2-methyltetrols and nighttime NO$_x$ in the main text:

*"In addition, nighttime 2-methyltetrol levels in the urban atmosphere deviate from the conventional understanding of isoprene SOA formation in terms of segregated NOx dependent regimes."*

43) **Lines 439-545:** "In this study ... (Riva et al., 2015)." I think the bulk of this text should go above in 3.3.2. with just a summary here - seems like you're presenting some new correlations here in the conclusions section.

We have already discussed these correlations in Section 3.3.2.

44) **Lines 555-556:** again, nearly invariant and ALWAYS very acidic is the key I think you're trying to present here.

We emphasized that aerosols are very acidic in this revised text on Page 27, Lines 603-607:

*"The absence of a correlation of aerosol acidity with MAE/HMML- and IEPOX-derived SOA tracers indicates that acidity is not the limiting variable that controls formation of these compounds. Because the aerosols are acidic (campaign average aerosol pH of 1.8), the lack of correlation between SOA tracers and acidity may stem from the nearly invariant aerosol acidity throughout the campaign."*

45) **Line 560:** "since urban emissions are directly present" => "in the presence of fresh urban emissions"

We agree with the referee's suggestion. We revised the sentences as follows on Page 27, Lines 611-612:

*"Future work should examine how well current models can predict the isoprene SOA levels observed during this study, especially in the presence of fresh urban emissions."*

46) **Table 1:** the periods for the intensive aren't clear to me- the 4 sampling periods suggest coverage of 2 days, but these periods list 3 days-?

The 4 sampling periods suggest coverage of 2 days, and these schedules are run for 3 days. Detailed examples of intensive periods during June 10 – 12 and regular (day/night) on June 13 are consecutively illustrated in this table to aid in understanding the sampling schedule:

| Sampling period | Sampling start | Sampling stop |
|---|---|---|
| Intensive 1 | 06/10/2013, 8 am | 06/10/2013, 12 pm |
| Intensive 2 | 06/10/2013, 1 pm | 06/10/2013, 3 pm |
| Intensive 3 | 06/10/2013, 4 pm | 06/10/2013, 7 pm |
| Intensive 4 | 06/10/2013, 8 pm | 06/11/2013, 7 am (next day) |
| Intensive 1 | 06/11/2013, 8 am | 06/11/2013, 12 pm |
| Intensive 2 | 06/11/2013, 1 pm | 06/11/2013, 3 pm |
| Intensive 3 | 06/11/2013, 4 pm | 06/11/2013, 7 pm |
| Intensive 4 | 06/11/2013, 8 pm | 06/12/2013, 7 am (next day) |
| Intensive 1 | 06/12/2013, 8 am | 06/12/2013, 12 pm |
| Intensive 2 | 06/12/2013, 1 pm | 06/12/2013, 3 pm |
| Intensive 3 | 06/12/2013, 4 pm | 06/12/2013, 7 pm |
| Intensive 4 | 06/12/2013, 8 pm | 06/13/2013, 7 am (next day) |
| Regular daytime | 06/13/2013, 8 am | 06/13/2013, 7 pm |
| Regular nighttime | 06/13/2013, 8 pm | 06/14/2013, 7 am (next day) |

This information is already summarized concisely in Table 1 of the main text.

47) **In table 4:** I assume the bold lines are aggregated tracers for MAE/HMML vs. IEPOX? Explain in the caption

We added the following footnotes in Table 4 as the referee suggested to explain this:

"*Summed tracers for MAE/HMML-derived SOA"

"**Summed tracers for IEPOX-derived SOA"

48) **Table5:** "average amount detected tracers" =>"average fraction of detected tracers"

We corrected the column titles in Table 5 as the referee suggested here.

49) **Fig.2:** Looks like $NO_x$ peaks are mostly during fires, based on CO spikes concurrent? Does this affect any of the plume age analysis? It's quite hard to discern any day/night patterns here - maybe average day & night values, with SD bars, for some key metrics would be a good figure to include in the main body text? Also, add your plume age calc here to the time

series? I'm curious how much it varies over the campaign vs. has a typical diurnal pattern.

CO correlated with $NO_x$ ($r^2 = 0.39$) suggesting the sources of combustion including fires and tailpipe emissions. This effect wasn't included in the plume age analysis in this study. The diurnal plots of key parameters in Figure 2 have been shown separately for better visibility in Figure 3-5 and Figure S4 in SI. The plume ages ($NO_x$:$NO_y$) were 0.37 – 1.02 over the course of sampling period. The authors decided not to include plume ages in time series plot for simplicity as plume age didn't provide major information for the analysis. However, we are providing the times series and diurnal plot here for the referee's information.

[Figure]

[Figure]

50) **In caption of Fig. 4:** say something about this being a smaller fraction than Fig.3 –because significantly weaker correlation.

We added the information to the caption as the referee suggested. Please note that the figure

number changed from 4 to 7 because we moved some figures from SI to the main body of the paper.

*"Figure 7. Correlation of IEPOX-derived SOA tracers with (a) daytime $NO_2$, (b) daytime $O_3$, (c) daytime $P[NO_3]$, (d) nighttime $NO_2$, (e) nighttime $O_3$, and (f) nighttime $P[NO_3]$. Nighttime $P[NO_3]$ correlation suggests that $NO_3$ radical chemistry could explain some fraction of the IEPOX-derived SOA tracer concentrations. The contribution of nighttime $P[NO_3]$ to IEPOX-derived SOA would be smaller than MAE/HMML-derived SOA due to the weaker correlation."*

**SI: I would put S5-S7 in the main body of the paper.** Also, in those captions, when you say there is "no significant variation was observed amongst intensive samples", do you mean to compare different date periods where you did the 4 time chunks, or do you mean between the 4 time chunks over all of the date periods where you did that finer time resolution, or both? Please clarify an SI figure with a couple key structures & corresponding acronyms would be nice (MAE,HMML,etc.)

Figures S5-S7 have been moved to the main body of the paper as the referee suggested. "No significant variation was observed amongst intensive samples" means between the time chunks over all of the date periods where we did that finer time resolution. All figures in SI have been revised for the key structures & corresponding acronyms.

**References**

[revised manuscript text omitted]

---

## Author Comment (AC2) · 2 Apr 2016

**General comments:**

The focus of this manuscript is on the relationships between isoprene SOA markers and anthropogenic parameters such as sulfate and $NO_x$ during a field campaign conducted in Birmingham (BHM), Alabama, an urban site in the southeastern USA, where regional isoprene emissions from deciduous trees and local anthropogenic emissions ($SO_2$ and $NO_x$) are substantial. The study reveals the complexity and potential multitude of chemical pathways leading to isoprene SOA formation. It is interesting to learn that ozone also plays a role in forming isoprene SOA. It is evident from this field study that the IEPOX-related SOA markers contributed by far the most, while the methacrolein-related tracers were rather minor. It is also shown that acidity is not a limiting factor for isoprene SOA formation at the BHM site. Understanding the formation mechanisms of biogenic SOA, especially with regard to anthropogenic emissions, is indeed important for the development of more accurate models that are needed for $PM_{2.5}$ control abatement strategies. I only have some specific issues (mostly minor), which would profit from clarification.

**Specific comments:**

**Lines 229-230:** The EIC of m/z 199 was used to quantify the MAE/HMML-derived OS. It has been quite well established that the major MAE-derived OS that is present in ambient fine aerosol is 3-sulfooxy-2-hydroxy-2-methyl propanoic acid with a terminal sulfooxy group (Gómez-Gonzálezet al.,2010). There is, to my knowledge, no chemical evidence for a HMML-derived OS containing a terminal hydroxymethylgroup. Therefore, it would be best to be more conservative here and write: "...... to quantify the MAE-derived OS,....."

Ref.: Gómez-González, Y., Surratt, J. D., Cuyckens, F., Szmigielski, R., Vermeylen, R., Jaoui, M., Lewandowski, M., Offenberg, J. H., Kleindienst, T. E., Edney, E. O., Blockhuys, F., Van Alsenoy, C., Maenhaut, W., and Claeys, M.: Characterization of organosulfates from the photooxidation of isoprene and unsaturated fatty acids in ambient aerosol using liquid chromatography/(-) electrosprayionization massspectrometry, J.MassSpectrom.,43,371–382,2008.

We agree that only MAE-derived OS can be quantified by the authentic MAE-derived OS synthesized in-house. Thus, we remove the term HMML in the experimental section.

We have revised the sentence as follows on Page 11, Lines 242-243:

*"EICs of m/z 215, 333 and 199 were used to quantify the IEPOX-derived OS, IEPOX-derived dimer OS and the MAE-derived OS, respectively (Surratt et al., 2007a)."*

We have also revised the sentence as follows on Page 11, Lines 247-249:

*"The MAE-derived OS was quantified using an authentic MAE-derived OS standard synthesized in-house by a procedure to be described in a forthcoming publication ($^1$H NMR trace, Figure S2).*

However, as described in the introduction section, recent studies (Surratt et al., 2006; Surratt et al., 2010; Lin et al., 2013a; Nguyen et al., 2015) have shown that isoprene is oxidized under high-$NO_x$ conditions to yield methacrolein, which is then further oxidized by OH radicals in the presence of $NO_2$ to yield methacryloylperoxynitrate (MPAN). When MPAN is oxidized by OH it

yields at least two SOA precursors, including methacrylic acid epoxide (MAE) and hydroxymethyl-methyl-α-lactone (HMML). MAE and HMML can yield 2-methylglyceric acid (2-MG) and its OS derivative (Lin et al., 2013a; Nguyen et al., 2015). As a result, the MAE-derived OS tracer we measured could likely be derived from both MAE and HMML, even though the existence for HMML has only been indirectly measured by Nguyen et al. (2015). As a result, we will continue using the terminology MAE/HMML-derived OS in the subsequent sections of the manuscript to remind readers about the potential contribution from both MAE and HMML pathways. To be clearer on this issue, we added the following sentences on Pages 11-12, Lines 249-256:

*"Although the MAE-derived OS (Gómez-González et al., 2008), which is more formally called 3-sulfooxy-2-hydroxy-2-methyl propanoic acid, has been chemically verified from the reactive uptake of MAE on wet acidic sulfate aerosol (Lin et al., 2013a), the term MAE/HMML-derived OS will be used hereafter to denote the two potential precursors (MAE and HMML) contributing to this OS derivative as recently discussed by Nguyen et al. (2015). It should be noted that Nguyen et al. (2015) provided indirect evidence for the possible existence of HMML. As a result, further work is needed to synthesize this compound to confirm its structure and likely role in SOA formation from isoprene oxidation."*

**Line 238:** Mention is also made here of glyoxal-, methylglyoxal and hydroxyacetone-derived OS, but earlier in the introduction nothing is said about the chemical pathways leading to these products. For completeness, I suggest to briefly provide some background information about these pathways.

We agree with the referee's comment. We have added the following information into the introduction section on Pages 5-6, Lines 111-117:

*"In addition to MACR, other key oxidation products of isoprene, including glycolaldehyde, methylglyoxal, and hydroxyacetone, can undergo multiphase chemistry to yield their respective OS derivatives (Olsen et al., 2011; Schindelka et al. 2013; Shalamzari et al., 2013; Noziere et al., 2015). However, the contribution of isoprene on the glyoxal-, methylglyoxal-, and hydroxyacetone-derived OS mass concentrations in the atmosphere remains unclear since these*

*SOA tracers can also be formed from a wide variety of biogenic and anthropogenic precursors (Galloway et al., 2009, Liao et al., 2015)."*

**Lines 335-337:** It is mentioned that the IEPOX-derived SOA tracers comprised 92.5% of the total detected SOA tracer mass. To arrive at this value the 2-methyltetrols should not be counted twice, since they are also formed from IEPOX OS in the analytical GC/MS procedure by hydrolysis of the corresponding OS. How was this value of 92.5% estimated?

We agree with the point suggested by the referee. Thus, we did additional quality control experiments to investigate the exact impact of the IEPOX-derived OS during the GC/MS procedure and analysis. We directly injected known concentrations (i.e., 1, 5, 10, and 25 ppmv) of the IEPOX-derived OS standard into the GC/MS following trimethysilylation. We found the signals of 2-methylthreitol and 2-methylerythritol derived from the GC/MS analysis of IEPOX-derived OS were small (1.69% and 2.42%, respectively). We corrected for this effect in the estimation of IEPOX-derived SOA to the total detected SOA tracer mass in our manuscript. The revised estimation is now 92.45%, which did not significantly change from the original value of 92.5%.

However, we decided to use the revised estimation and added this information in the main text as follows in Section 2.2.2, Pages 10-11, Lines 226-231:

*"To investigate the effect of IEPOX-derived OS hydrolysis/decomposition during GC/EI-MS analysis, known concentrations (i.e., 1, 5, 10, and 25 pppv) of the authentic IEPOX-derived OS standard (Budisulistiorini et al., 2015)) were directly injected into the GC/MS following trimethylsilylation. Ratios of detected 2-methyltetrols to the IEPOX-derived OS were applied to estimate the total IEPOX-derived SOA tracers in order to avoid double counting when combining the GC/MS and UPLC/ESI-HR-QTOFMS SOA tracer results."*

We also added the following clarifying information in Section 3.2, Pages 16-17, Lines 357-363:

*"Our investigation for the potential of OS hydrolysis/decomposition during GC/EI-MS analysis*

*demonstrated that only 1.7% of 2-methylthreitol and 2.4% of 2-methylerythritol could be derived from the IEPOX-derived OSs. In order to accurately estimate the mass concentrations of the IEPOX-derived SOA tracers, we took this effect into account. Together, the IEPOX-derived SOA tracers, which represent SOA formation from isoprene oxidation predominantly under the low-NOx pathway, comprised 92.45% of the total detected isoprene-derived SOA tracer mass at the BHM site.*"

**Line 340:** See comment about MAE/HMML-OS above.

As described above, we decided to use the terminology "MAE/HMML-OS" in the Results and Discussion section to emphasize its formation pathway. We did this as follows on Pages 11-12, Lines 249-256:

*"Although the MAE-derived OS (Gómez-González et al., 2010), which is more formally called 3-sulfooxy-2-hydroxy-2-methyl propanoic acid, has been chemically verified from the reactive uptake of MAE on wet acidic sulfate aerosol (Lin et al., 2013a), the term MAE/HMML-derived OS will be used hereafter to denote the two potential precursors (MAE and HMML) contributing to this OS derivative as recently discussed by Nguyen et al. (2015). It should be noted that Nguyen et al. (2015) provided indirect evidence for the possible existence of HMML. As a result, further work is needed to synthesize this compound to confirm its structure and likely role in SOA formation from isoprene oxidation.*"

**Lines 359-361:** It is not clear what the authors want to say by writing: "The low isoprene SOA/OM ratio is consistent with the low WSOC/OC reported in section 3.1, suggesting an increased contribution of primary OA or secondary OM to the total OM at BHM". Some clarification should be given about the secondary OM; perhaps this should be better described as "hydrophobic secondary OM", originating from anthropogenic emissions.

We agree with the referee. The sentence has been revised as follows on Pages 17-18, Lines 385-387:

*"The low isoprene SOA/OM ratio is consistent with the low WSOC/OC reported in Section 3.1, suggesting an increased contribution of primary OA or hydrophobic secondary OM originating*

*from anthropogenic emissions to the total OM at BHM."*

**Lines 386-408:** Throughout this section, MAE/HMMLOS is mentioned several times. See comment about MAE/HMML-OS above.

Since this comment was already raised above by the reviewer, please refer to our replies above in how we exactly dealt with this reviewer comment.

**Section 3.3.2** Effect of $O_3$: It is very interesting to learn that $O_3$ has an effect on 2-methyltetrol formation, but not on C5-alkene triol formation, consistent with a recent study by Riva et al. (2016). What I am missing here is some brief mechanistic information about 2-methyltetrol formation from isoprene by the action of $O_3$, so that the interested reader does not have to look up the original article. I am also curious to know whether there was an intercorrelation between the 2-methyltetrols and the C5-alkenetriols; a strong intercorrelation would support that they both are formed through the IEPOX multiphase pathway.

We agree with the referee's comment. We have added brief mechanistic information about 2-methyltetrol formations from isoprene ozonolysis as follows on Pages 22-23, Lines 491-501:

*"Previous studies (Nguyen et al., 2010; Inomata et al., 2014) proposed that SOA formation from isoprene ozonolysis occurs from stabilized Criegee intermediates (sCIs) that can further react in the gas phase to form higher molecular weight products that subsequently partition to the aerosol phase to make SOA. Recent work by Riva et al. (2016) systematically demonstrated that isoprene ozonolysis in the presence of wet acidic aerosol yields 2-methyltetrols and organosulfates unique to this process. Notably, no C5-alkene triols were observed, which are known to form simultaneously with 2-methyltetrols if IEPOX multiphase chemistry is involved (Lin et al., 2012). Riva et al. (2016) tentatively proposed that hydroperoxides formed in the gas phase from isoprene ozonolysis potentially partition to wet acidic sulfate aerosols and hydrolyze to yield 2-methyltetrols as well as the unique set of organosulfates observed (Riva et al., 2016). Additional work using authentic hydroperoxide standards is needed to validate this tentative hypothesis."*

Regarding an intercorrelation between the 2-methyltetrols and the C5-alkene triols, we observed

a strong correlation ($R^2$ = 0.84) during nighttime suggesting that they both are formed through the IEPOX multiphase pathway. However, a mild correlation ($R^2$ = 0.55) is observed during daytime suggesting that $O_3$ may contribute some fraction on 2-methyltetrols formation which is in agreement with the key finding from isoprene ozonolysis by Riva et al., 2016.

**Technical corrections:**

**Line 35:** ...... coupled to electrospray ionization high-resolution quadrupole time-of-flight massspectrometry.....

We made the correction as the referee suggested on Page 2, Lines 33-37:

*"Sample extracts were analyzed by gas chromatography/electron ionization-mass spectrometry (GC/EI-MS) with prior trimethylsilylation and ultra performance liquid chromatography coupled to electrospray ionization high-resolution quadrupole time-of-flight mass spectrometry (UPLC/ESI-HR-QTOFMS) to identify known isoprene SOA tracers."*

**Line 148:**.....to measure known.......or.......to determine quantities of...... .

We made the correction as the referee suggested on Page 7, Lines 152 - 156:

*"The analysis of $PM_{2.5}$ was conducted in order to determine quantities of known isoprene SOA tracers and using collocated air quality and meteorological measurements to investigate how anthropogenic pollutants including $NO_x$, $SO_2$, aerosol acidity (pH), $PM_{2.5}$ sulfate ($SO_4^{2-}$), and $O_3$ affect isoprene SOA formation."*

**Line 176:**...... and $NO_x$ were forecast by......

We made the correction as the referee suggested on Pages 8-9, Lines 180-184:

*"The intensive sampling schedule was conducted on days when high levels of isoprene, $SO_4^{2-}$ and $NO_x$ were forecast by the National Center for Atmospheric Research (NCAR) using the Flexible Particle dispersion model (FLEXPART) (Stohl et al., 2005) and Model for Ozone and Related Chemical Tracers (MOZART) (Emmons et al., 2010) simulations."*

**Line 254:**......  baked at 500 degrees C for.....(space before degrees C)

We made the correction as the referee suggested on Pages 12-13, Lines 273-275:

 *"To maintain low background carbon levels, all glassware used was washed with water, soaked in 10% nitric acid, and baked at 500 ℃ for 5 h and 30 min prior to use."*

**Line 258:** The abbreviation "PTFE" needs to be introduced here.

We made the correction as the referee suggested on Page 13, Lines 278-279:

 *"Extracts were then passed through a 0.45 µm polytetrafluorethylene (PTFE) filter to remove insoluble particles."*

**Line 352:**......  based on recent studies.....

We made the correction as the referee suggested on Page 17, Lines 377-379:

 *"Isoprene SOA contribution to total OM was estimated by assuming the OM/OC ratio 1.6 based on recent studies (El-Zanan et al., 2009; Simon et al., 2011; Ruthenburg et al., 2014; Blanchard et al., 2015)."*

**Line 419:**.....$NO_3$-initiated oxidation......

We made the correction as the referee suggested on Page 21, Lines 457-459:

 *"However, Schwantes et al. (2015) demonstrated that $NO_3$-initiated oxidation of isoprene yields isoprene nitrooxy hydroperoxides (INEs) through nighttime reaction of $RO_2 + HO_2$, which upon further oxidation yielded isoprene nitrooxy hydroxyepoxides (INHEs)."*

**References**: titles of journal articles should not be capitalized; some still are capitalized and should be corrected.

We made the correction as the referee suggested in the reference section.

**Line 807:**......  trans-3-methyl-3,4-dihydroxytetrahydrofuran,.....

We made the correction as the referee suggested on Pages 37-38, Lines 884-887:

*"Zhang, Z., Lin, Y.-H., Zhang, H., Surratt, J., Ball, L., and Gold, A.: Technical Note: Synthesis of isoprene atmospheric oxidation products: isomeric epoxydiols and the rearrangement products cis-and trans-3-methyl-3,4-dihydroxytetrahydrofuran, Atmos. Chem. Phys., 12, 8529-8535, 2012"*

**Figure 2–panel (d):** the trace for MAE-OS in light purple/rose color is hardly visible.

We made the correction as the referee suggested for Figure 2 panel (d). The colors in panel (d) were improved for better visibility.

**References**

[revised manuscript text omitted]